# Genome-wide cross-cancer analysis illustrates the critical role of bimodal miRNA in patient survival and drug responses to PI3K inhibitors

**Laura Moody[1], Guanying Bianca Xu[2], Yuan-Xiang Pan[1,2,3], Hong Chen[1,2]***

**1** Division of Nutritional Sciences, University of Illinois at Urbana-Champaign, Urbana, Illinois, United States of America, **2** Department of Food Science and Human Nutrition, University of Illinois at Urbana-Champaign, Urbana, Illinois, United States of America, **3** Illinois Informatics Institute, University of Illinois at Urbana-Champaign, Urbana, Illinois, United States of America

* hongchen@illinois.edu

**Data Availability Statement:** Codes used to calculate bimodality index, to generate histograms, and to plot survival curves have been deposited in

## Abstract

Heterogeneity of cancer means many tumorigenic genes are only aberrantly expressed in a subset of patients and thus follow a bimodal distribution, having two modes of expression within a single population. Traditional statistical techniques that compare sample means between cancer patients and healthy controls fail to detect bimodally expressed genes. We utilize a mixture modeling approach to identify bimodal microRNA (miRNA) across cancers, find consistent sources of heterogeneity, and identify potential oncogenic miRNA that may be used to guide personalized therapies. Pathway analysis was conducted using target genes of the bimodal miRNA to identify potential functional implications in cancer. *In vivo* overexpression experiments were conducted to elucidate the clinical importance of bimodal miRNA in chemotherapy treatments. In nine types of cancer, tumors consistently displayed greater bimodality than normal tissue. Specifically, in liver and lung cancers, high expression of miR-105 and miR-767 was indicative of poor prognosis. Functional pathway analysis identified target genes of miR-105 and miR-767 enriched in the phosphoinositide-3-kinase (PI3K) pathway, and analysis of over 200 cancer drugs *in vitro* showed that drugs targeting the same pathway had greater efficacy in cell lines with high miR-105 and miR-767 levels. Overexpression of the two miRNA facilitated response to PI3K inhibitor treatment. We demonstrate that while cancer is marked by considerable genetic heterogeneity, there is between-cancer concordance regarding the particular miRNA that are more variable. Bimodal miRNA are ideal biomarkers that can be used to stratify patients for prognosis and drug response in certain types of cancer.

## Author summary

Bimodal genes can be defined as those having two modes of expression within the same population. A variety of statistical methodologies have been employed to assess bimodal gene expression, but current methods and their applications have been limited. Given the advances in next-generation sequencing as well as the extensive regulatory role of

GitHub (https://github.com/nutrigenelab/
bimodality).

**Funding:** The research is supported by the
following grants: grants from the USDA
Cooperative State Research, Education and
Extension Service (Hatch project numbers # ILLU-
971-344 and ILLU-698-369) to HC, a grant from
Cancer Scholars for Translational and Applied
Research (C*STAR) program from Carle
Foundation Hospital, and the Office of the Vice
Chancellor for Research in University of Illinois at
Urbana-Champaign to LM and YXP. The funders
had no role in study design, data collection and
analysis, decision to publish, or preparation of the
manuscript.

**Competing interests:** The authors have declared
that no competing interests exist.

miRNA, assessing bimodality in miRNA-seq data can greatly broaden our understanding
of factors underlying tumor progression. The goal of the current study was to utilize a
novel mixture modeling approach to identify bimodal miRNA and then demonstrate
their importance in cancer by evaluating their ability to predict overall survival and drug
response. Our results showed that high levels of bimodal miRNA expression was charac-
teristic of cancer. Additionally, several bimodal miRNA were common to multiple cancer
types, suggesting that certain miRNA consistently account for tumor heterogeneity and
may be involved in general oncogenic processes. Our study points to the potential of
bimodal miRNA to facilitate precise prognostic evaluation and effective treatment
strategies.

## Introduction

Cancer is classically characterized by genomic instability and mutations which drive uncon-
trolled cellular growth, heightened angiogenesis and metastasis, and metabolic abnormalities
[1]. However, mutational and transcriptomic profiles are tumor-specific, resulting in a high
degree of heterogeneity among cancer patients. Not only is there considerable heterogeneity
between cancer types, but even two tumors within the same cancer type and stage often display
very different genetic profiles. Transcriptomic heterogeneity is exemplified by bimodal gene
expression. Bimodal genes can be defined as those having two modes of expression within the
same population. Bimodal genes act as molecular switches that define cancer subtypes. One
example of bimodal gene expression is the estrogen receptor (*ESR1*) in breast cancer. In
regards to *ESR1* expression, breast tumors have one of the two molecular subtypes, one that
expresses *ESR1* and the other shuts down expression (ER-). These discoveries have been partic-
ularly informative not only in prognostic prediction but also in guiding treatment regimens
and understanding the efficacy of hormone-based therapies and drugs that target specific
receptors [2–4]. Thus, bimodal genes represent a set of tumorigenic genes which can motivate
effective therapeutics.

A variety of statistical methodologies have been employed to assess bimodal gene expres-
sion, including test statistics that reflect significant outliers and spacing of data as well as heu-
ristic clustering [5–7]. Model-based clustering has been an effective tool for evaluating
bimodality. Teschendorff et al. proposed mixture modeling as a method to define major sub-
groups of a population [8]. First, the expectation-maximization (EM) algorithm was used to
estimate the parameters of a Gaussian mixture model and then the Bayesian information crite-
rion (BIC) was employed to choose between unimodal and bimodal models. Next, genes were
further filtered based on kurtosis. This method was modified by Wang et al. by using Markov
chain Monte Carlo methods instead of the EM algorithm [9]. As opposed to kurtosis, the
bimodality index (BI) was also introduced as a metric which considers the mean and propor-
tion of observations within each cluster to rank genes based on bimodality.

While mixture modeling has been pivotal for identifying and understanding bimodal
genes, current methods and their applications have been limited. First, current methods do
not take into account the underlying distribution of a gene and are thus prone to false posi-
tives. There is no method that compares the gene expression distributions in tumor and con-
trol tissue. Thus, genes that are bimodally expressed in healthy tissue and unlikely contribute
to tumorigenesis are not controlled. Additionally, only a handful of studies have applied mix-
ture modeling to assess high-throughput sequencing data [10] and microRNA (miRNA)
expression [11]. Given the advances in next-generation sequencing as well as the extensive

regulatory role of miRNA, assessing bimodality in miRNA-seq data can greatly broaden our understanding of factors underlying tumor progression.

The goal of the current study was to utilize a novel mixture modeling approach to identify bimodal miRNA (Fig 1A) and then demonstrate their importance in cancer by evaluating their ability to predict overall survival and drug response (Fig 1B). First, we analyzed miRNA-seq data across nine types of cancer and quantified bimodal miRNA to characterize oncogenic miRNA that could represent a novel set of clinically-relevant tumor biomarkers. Specifically, we used bimodal miRNA to stratify patients into two distinct categories and examined group differences in survival and drug response. In identifying important tumorigenic miRNA, we not only provide foundational insight into the major miRNA contributors to tumor variability, but also point to the potential of bimodal miRNA to facilitate precise prognostic evaluation and effective treatment strategies.

## Results

### Controlled mixture modeling validation

We first sought to validate a method that could reliably identify bimodal expression. In order to reduce false positives and focus only on genes that are relevant to cancer, bimodality was assessed using model-based clustering in both the cancer and control samples. Normalized RNA-seq data from breast tumors and non-tumor mammary tissue was downloaded from Genomic Data Commons (GDC, formerly TCGA). The data was then log2 transformed for further analysis. All stages were included in the analysis for a total of 1,102 tumor samples and 113 non-tumor samples.

For each of the 60,483 genes, we first performed mixture modeling on the tumor samples to decide whether a one-component or two-component model was the better fit. From the genes that were better fit by a two-component model, k-means was then used to re-cluster the tumor samples. The same steps were performed on the control samples. Finally, genes were ranked using the calculation of the bimodality index. If control samples were bimodally distributed, a penalty was imposed in the calculation of the bimodality index (see methods). This yielded a method that we refer to as controlled mixture modeling (CM). In order to show that the addition of control samples was beneficial, we compared our CM model to a mixture model without controls (MM). For each model, the bimodality index was calculated and genes were ranked highest to lowest.

First, we visually inspected the top bimodal genes identified using MM and CM. MM and CM both tended to favor genes in which a subset of patients had no expression and a second subset of showed higher expression (Fig 2A and 2B). MM and CM methods identified similar top three bimodal genes. The top three genes using MM were *SCGB2A2* (BI = 2.20), *SCGB1D2* (BI = 2.03), and *TFF1* (BI = 1.92; Fig 2A). The top bimodal gene using CM was *TFF* (BI = 1.92), and the second was a lncRNA (BI = 1.81). CM also identified *SCGB2A2* but the BI was penalized; thus the gene was ranked third (BI = 1.81; Fig 2B).

As expected, the addition of control samples reduced the number of genes at all bimodality index thresholds (Table 1). We hypothesized that by filtering genes using control samples, we could reduce false positives and focus on important tumorigenic genes. To test this hypothesis, we examined known bimodal genes that were relevant to disease progression in breast cancer, including *ESR1*, *HER2*, and progesterone receptor (*PGR*) [9]. The use of control samples was particularly beneficial in identifying bimodality in *ESR1* and *PGR*. CM ranked *ESR1* 25th compared to MM which ranked it 47th (Table 1). Our algorithm also ranked *PGR* 62nd. This was better than MM which ranked it 137th. Interestingly, *HER2* was not ranked highly by either

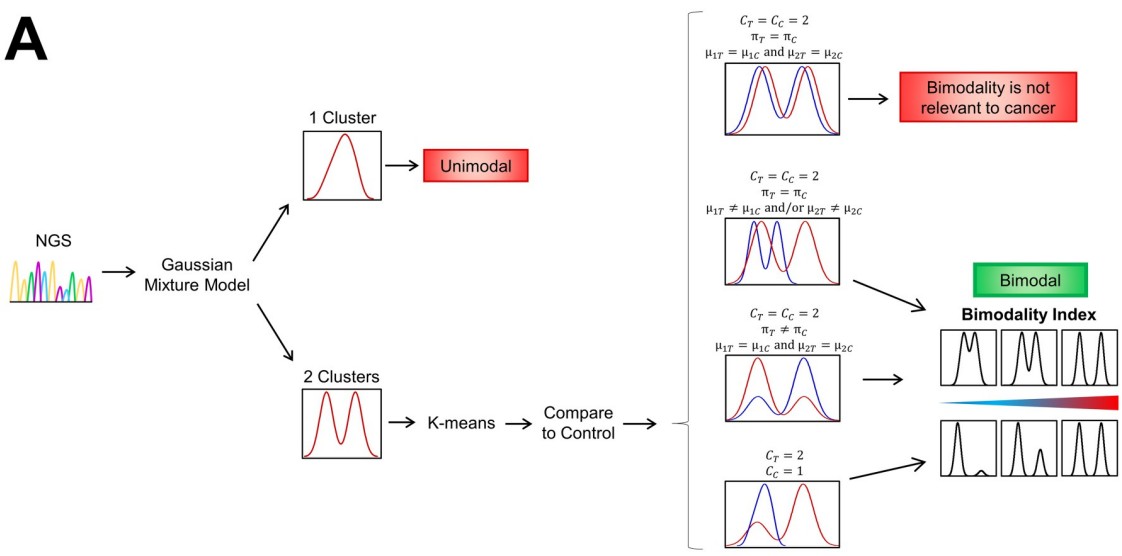

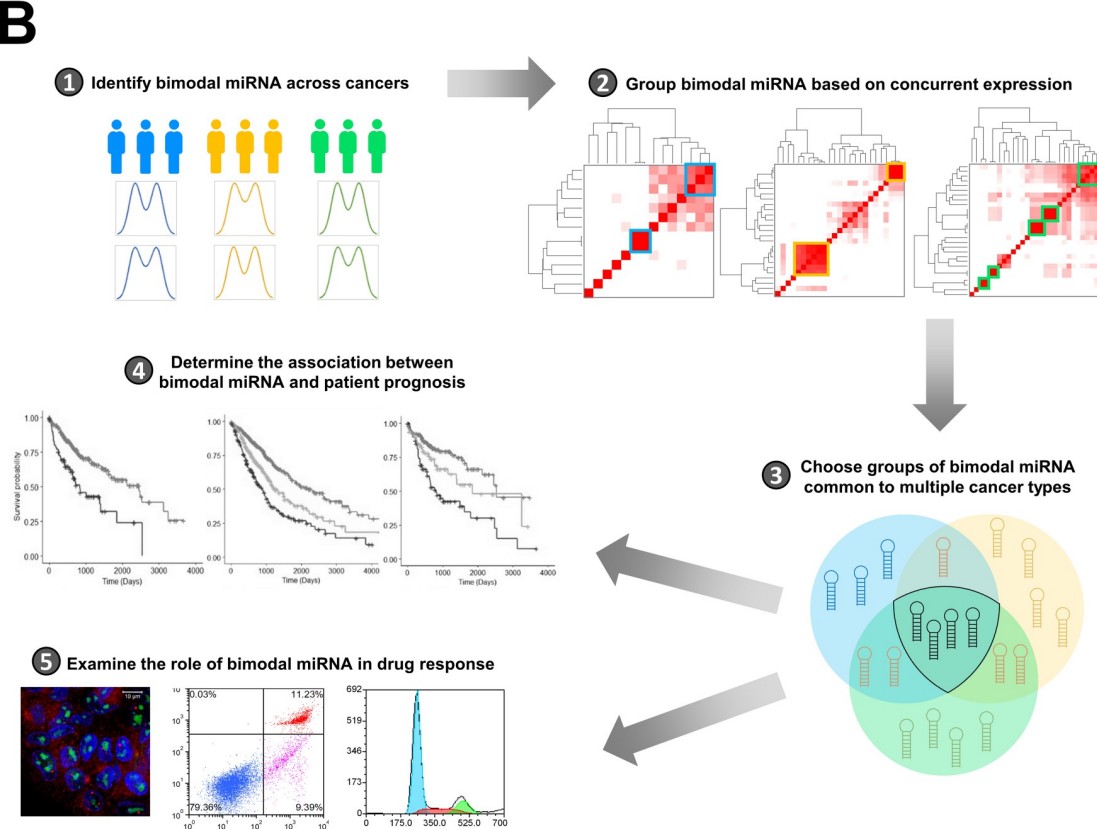

**Fig 1. Strategies for identification of bimodal miRNA and evaluation of their clinical relevance.** (A) Controlled mixture modeling (CM) was employed to identify bimodal miRNA. Gaussian mixture modeling was performed to identify one- and two-component miRNA. The distributions of two-component miRNA were then compared to control tissue before calculation of the bimodality index. (B) To assess the clinical relevance of bimodal miRNA, CM was used to identify bimodal miRNA across nine cancer types. Groups of concurrently expressed miRNA that were common to multiple cancer types were then further investigated for their role in patient prognosis and drug response.

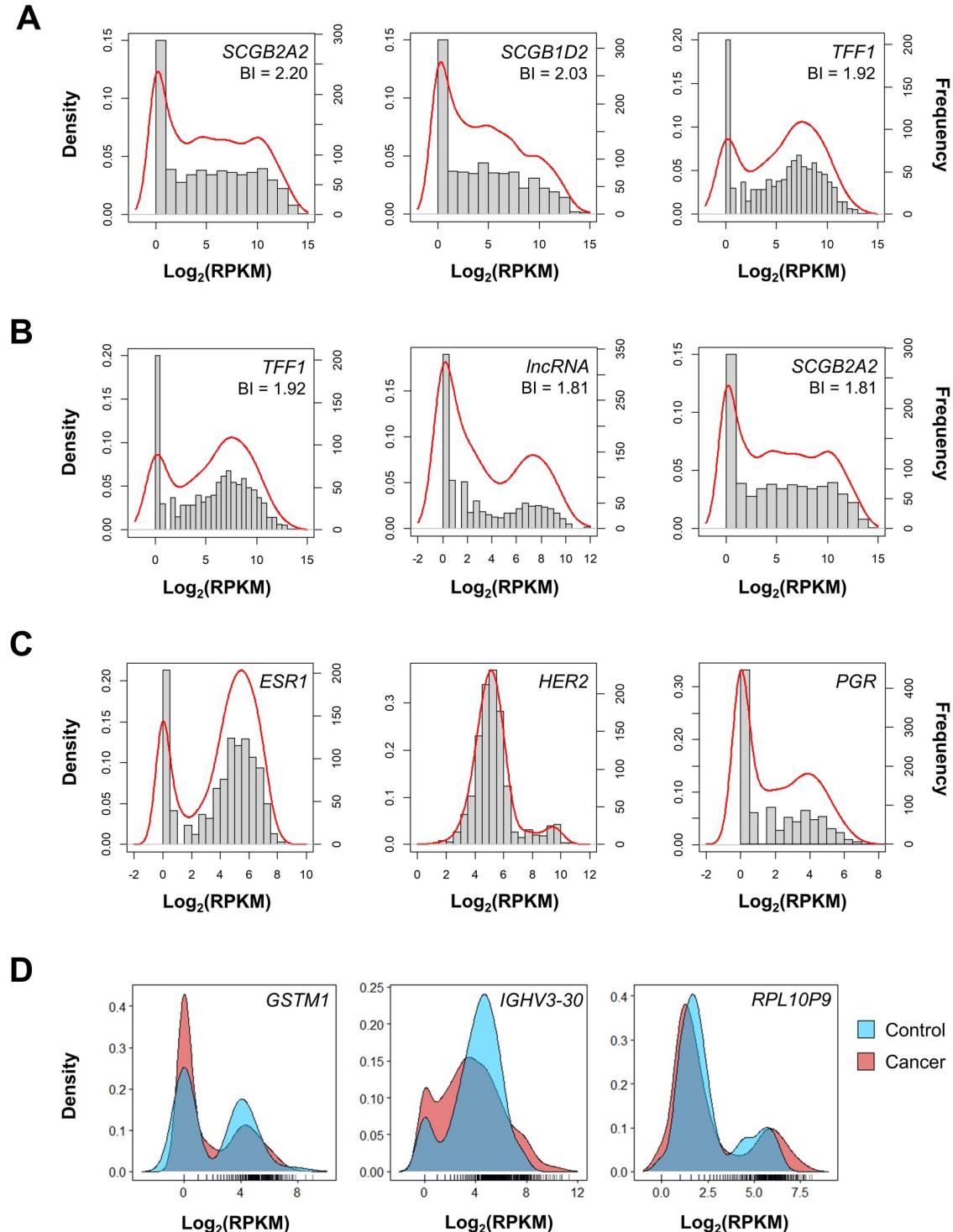

**Fig 2. Methodology validation.** Breast cancer RNA-seq data were clustered using (A) mixture modeling (MM) and (B) controlled mixture modeling (CM). The bimodality index was calculated and the top three genes using each method are reported. (C) *ESR1*, *HER2*, and *PGR* are known to follow a bimodal distribution among breast cancer patients. The plots show gene expression in log$_2$(RPKM) on the x-axis. Density is represented by the red line (left y-axis) and frequency is represented by the gray bars (right y-axis). (D) CM excluded genes with similar distributions in cancer and control. *GSTM1*, *IGHV3-30*, and *RPL10P9* were identified by MM but excluded by CM at BI > 1. Density is shown for cancer (red) and control (blue) samples.

**Table 1. Methodology performance.** Methods were tested on breast cancer RNA-seq data and compared with regards to the number of genes identified at various bimodality index (BI) thresholds and ranking of known bimodally expressed genes (*ESR1*, *HER2*, and *PGR*). CM: controlled mixture modeling, MM: mixture modeling.

| Parameter | MM | CM |
|---|---|---|
| Number with BI > 0 | 12417 | 6694 |
| Number with BI > 1 | 188 | 88 |
| Number with BI > 1.4 | 26 | 13 |
| Number with BI > 1.5 | 14 | 7 |
| Number with BI > 2.0 | 2 | 0 |
| BI rank of *ESR1* | 47 | 25 |
| BI rank of *HER2* | 471 | 503 |
| BI rank of *PGR* | 137 | 62 |

algorithm, but upon closer examination of the distribution, we found that in this particular dataset the *HER2* distribution that was rather ambiguous (Fig 2C).

Additionally, we visualized genes that were identified by MM but excluded by CM at a BI > 1. These genes had distributions that were very similar between cancer and control (Fig 2D). Of the 100 genes with BI > 1 that were identified by MM but excluded by CM, 33 coded for immunoglobulin chains. This finding is reflective of the high sequence variation observed in these genes within the general population [12]. Another gene excluded by CM at BI > 1 was glutathione S-transferase mu 1 (*GSTM1*) which is known to have variable enzymatic activity in the general population due to polymorphic deletions [13].

## Bimodal miRNA across cancers

After validating our methodology on breast cancer RNA-seq data, we applied the model to find bimodal miRNA. First, we downloaded miRNA-seq data from GDC for nine types of cancer and normal tissue. Due to data availability, we narrowed our investigation to cancers that had at least 30 non-tumor control samples. Each tissue type was individually assessed. We applied the MM approach on cancer and control tissue using bootstrapped samples of same size as normal tissue in order to account for differences in sample size. We found that cancer tissue had more bimodal miRNA than control when considering all sources of bimodality (bimodality index > 0, Fig 3A). This trend was less clear at higher BI thresholds (S1 Fig). Next, we looked at bimodality in cancer samples using CM. We set the bimodality index threshold to 1.2, 1.3, 1.4, and 1.5 and saw a decrease in the number of bimodal miRNA (Fig 3B). At a bimodality index threshold of 1.4, kidney cancer had the most bimodally expressed miRNA (n = 56), while prostate and breast cancer had the least (n = 11 and n = 14, respectively; Fig 3C). This trend was also observed at thresholds of 1.3 and 1.5. Additionally, our findings supported the results from the methodology validation by showing that the top-ranked miRNA from each cancer type had fairly evenly partitioned components and non-trivial expression, rather than miRNA with very few outliers (Fig 3D).

Next, we examined the concurrent expression of bimodal miRNA. We postulated that modules of concurrently expressed miRNA would have a greater functional impact than single miRNA. Pearson correlation coefficients were calculated between miRNA pairs. Only miRNA with bimodality indices > 1.4 were considered, as previous studies have shown that a bimodality index threshold of 1.4 returns a 0–1.2% false positive rate using a sample size of 50 and a 0% false positive rate when the sample size is ≥ 200 [9]. Groups of at least three miRNA were considered to be a module if all pairwise correlations were greater than 0.5. Size and number

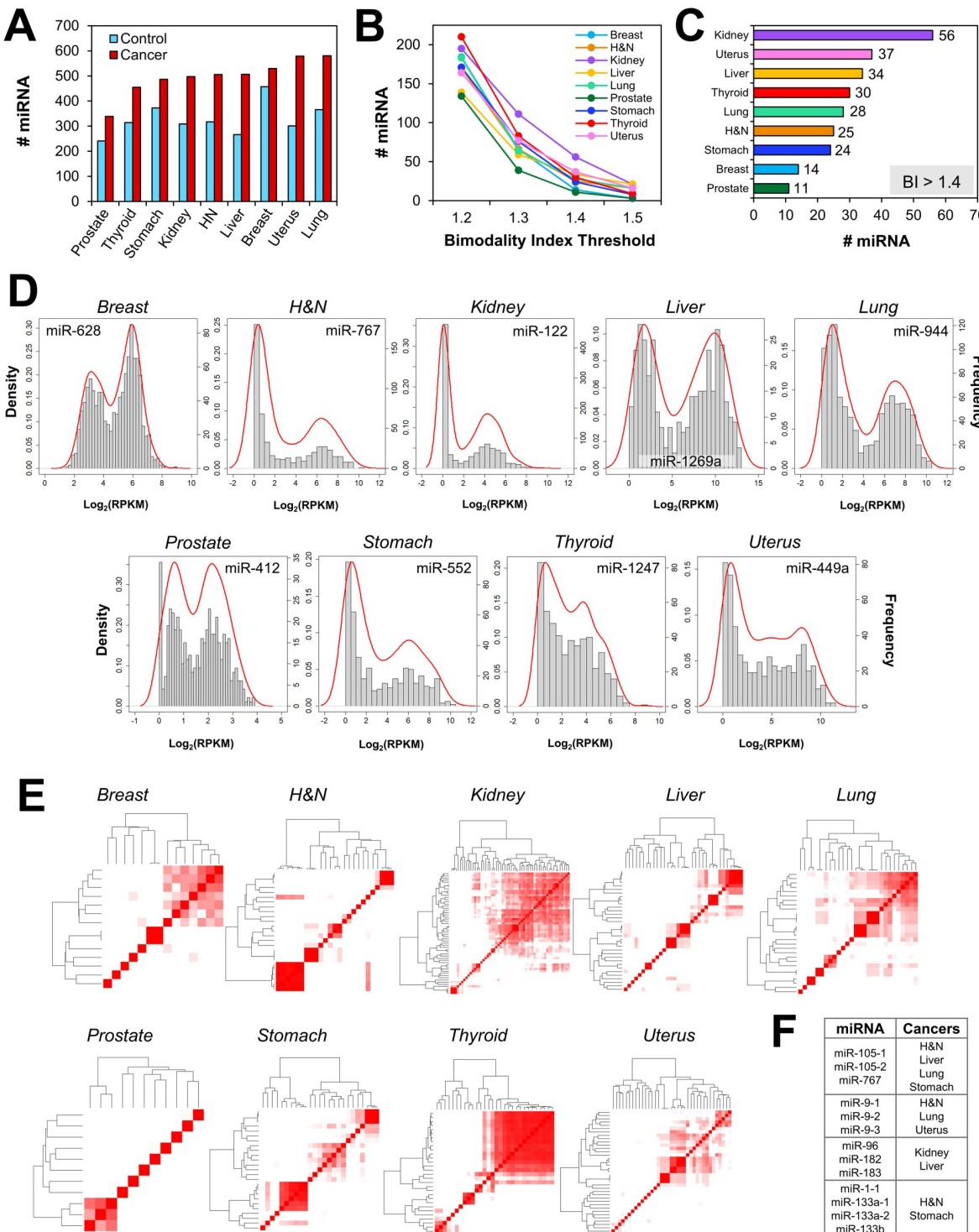

**Fig 3. Bimodal miRNA across cancers.** (A) Bimodally expressed miRNA were examined in nine types of cancer and control tissue using mixture modeling (MM). Bimodality was also examined in cancer tissue using controlled mixture modeling (CM), and (B) the number of bimodal miRNA at various bimodality index thresholds are reported, as well as (C) a more detailed report of bimodal miRNA having a bimodality index (BI) > 1.4. (D) The top-ranked bimodal miRNA from each cancer type were visually inspected for bimodality. (E) Within each cancer type, correlations were calculated between bimodal miRNA in order to identify modules of concurrently expressed bimodal miRNA. High correlations are represented in red. (F) Four concurrent expression modules were common to multiple cancer types.

of co-expression modules varied between cancers (Fig 3E). Kidney cancer had the largest number of co-expression modules (n = 5) containing a total of 31 miRNA (S1 Table). The largest module contained 14 miRNA and was observed in thyroid cancer. Liver, lung, stomach, uterus, and head and neck cancers had moderately sized modules containing three to six miRNA each. Additionally, prostate cancer had no modules, suggesting that its 11 bimodal miRNA were expressed independently of each other. Bimodal modules for each cancer are listed in S1 Table.

We then focused on modules of bimodal miRNA that were common across cancers. Just as certain oncogenes and tumor suppressor genes widely contribute to tumorigenesis (e.g. p53, KRAS, PIK3CA, etc.), we hypothesized that particular miRNA broadly regulate cancer progression. Thus, we found four modules that were common to multiple cancer types (Fig 3F). The most common module was found in head and neck, liver, lung, and stomach cancer and consisted of miR-105-1, miR-105-2, and miR-767. The second most common module consisted of miR-9-1, miR-9-2, and miR-9-3 and was present in head and neck, lung, and uterine cancer. miR-96, miR-182, and miR-183 were bimodal modules in kidney and liver cancer. Head and neck and stomach cancer also shared the bimodal expression of miR-1-1, miR-133a-1, miR-133a-2, and miR-133b.

## Survival analysis

After identifying bimodal miRNA, we next investigated whether they could be used to predict patient outcome. We first examined whether multiple miRNA modules were sufficient to stratify patients based on overall survival. Using all of the co-expression modules within each cancer type, hierarchical clustering was performed using Manhattan distance as the dissimilarity measure and complete linkage. Dendrograms were cut to yield two groups (Fig 4A–4H). Using the two groups, a Cox proportional hazards regression was fit and hazard ratio (HR) and median survival time were calculated (Fig 4I–4P, S2 Table). Overall, we found that multiple modules of concurrently expressed miRNA were successful at predicting patient outcome in head and neck (HR: 0.54, 95% CI: 0.41–0.70, p = 5.21 x $10^{-4}$), kidney (HR: 0.73, 95% CI: 0.56–0.96, p = 0.023), liver (HR: 0.46, 95% CI: 0.32–0.65, p = 1.10 x $10^{-5}$), thyroid (HR: 0.32, 95% CI: 0.21–0.86, p = 0.025), and uterine cancer (HR: 0.62, 95% CI: 0.40–0.97, p = 0.034). Additionally, there was a trend for significance in stomach cancer (HR: 0.74, 95% CI: 0.54–1.03, p = 0.077).

We then sought to determine whether single modules of bimodal miRNA could also be used to stratify patients based on overall survival. For this analysis, we used modules that were common to multiple cancer types. The most common co-expression module included miR-105-1, miR-105-2, and miR-767. These three miRNA are located on the X chromosome in the first intron of GABA A receptor alpha 3 (*GABRA3*) (Fig 5A). Patients were divided into high and low expressers. Wald test showed a significant difference in survival time between groups in liver cancer, such that higher miR-105 and miR-767 expression resulted in an HR of 2.37 compared to low expression (95% CI: 1.61–3.48, p < 0.001; Fig 5B). In lung cancer, this trend was also observed (HR: 1.26, 95% CI: 1.03–1.56, p = 0.028; Fig 5C). In head and neck and stomach cancer, there was no difference between high and low miRNA expression groups (p = 0.71 and p = 0.52, respectively; S1 Fig). We also looked at the tumor node metastasis (TNM) staging of both high- and low-expressing patients and compared distributions using Chi square tests. In liver cancer, each TNM stage contained even numbers of high- and low-expressing individuals, suggesting that stage and miRNA provide two distinct stratifications (p = 0.28; Fig 5D). In lung cancer, high expressers tended to fall into stage III (p = 0.021). These findings were supported by *in vitro* experimentation in liver cancer (HepG2) and lung

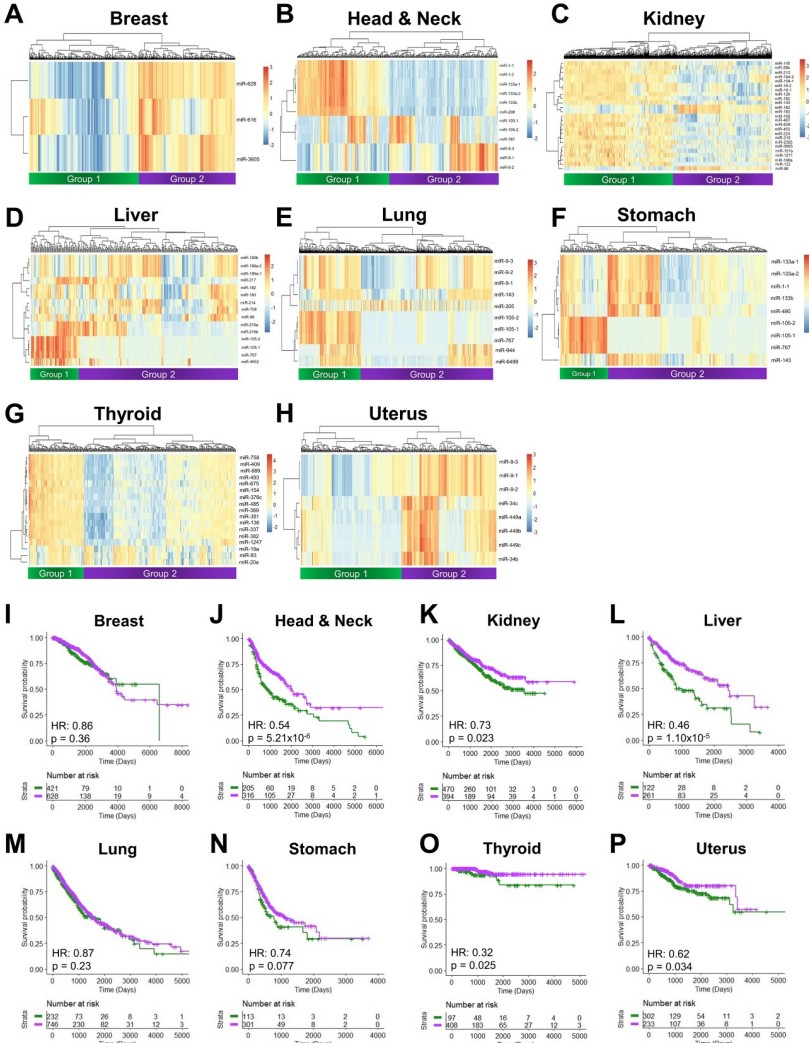

**Fig 4. Patient stratification and survival analysis using bimodal miRNA modules.** (A-H) Hierarchical clustering was performed using bimodal miRNA co-expression modules, and patients were divided into two groups. (I-P) Using the two groups, a Cox proportional hazards regression was fit for each cancer type. The reported hazard ratio (HR) denotes the risk of death in Group 2 (purple) compared to Group 1 (green). The statistical significance (p-value) of each HR was determined using a Wald test.

cancer (A549) cell lines. Cells were treated with miR-105 and miR-767 mimics, and cell viability was measured via WST-1 assay. A 74% increase in cell viability was observed in HepG2 cells, and a 20% increase in cell viability was observed in A549 cells (Fig 5E). Furthermore, miRNA-treated HepG2 and A549 cells also had elevated expression of the cellular proliferation marker, Ki-67 (Fig 5F and 5G).

The three other miRNA modules that were common to multiple cancer types were investigated for their ability to predict patient survival. Notably, high miR-9 was associated with shorter survival in head and neck cancer (HR: 0.72, 95% CI: 0.53–0.97, p = 0.033), but not in lung and uterine cancers (p = 0.31 and p = 0.36, respectively; S3 Fig). High miR-96, miR-182, and miR-183 were beneficial to overall survival in kidney cancer (HR: 0.65, 95% CI: 0.49–0.87, p = 0.003), but not in liver cancer (p = 0.85; S4 Fig). Head and neck and stomach cancer shared

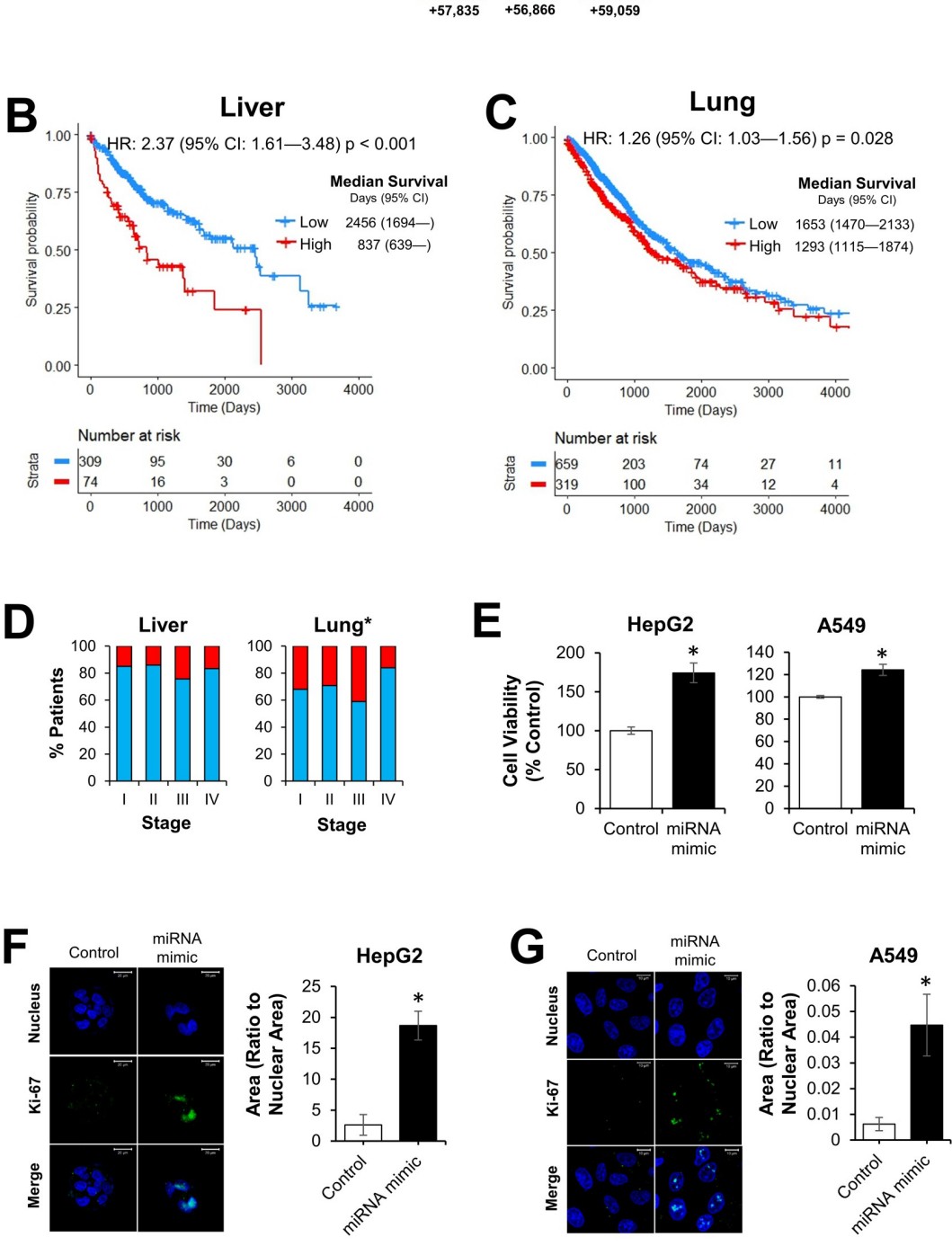

**Fig 5. miR-105 and miR-767 are associated with overall survival and cellular proliferation in liver and lung cancer.** (A) genomic location of miR-105-1, miR-105-2, and miR-767. (B) Liver and (C) lung cancer patients were divided into low and high miRNA expressers. For each cancer type, a Cox proportional hazards regression was fit. The reported hazard ratio (HR) and 95% confidence interval (CI) denote the risk of death in the high expression group compared to the low expression group. Median survival time is given in days with a 95% CI. (D) The relationship between miRNA expression and TNM stage. Graphs show the proportion of individuals with high (red) and low (blue) miR-105 and miR-767 expression within each stage. Differences in TNM staging between high and low expressers was tested using Chi square tests. *p<0.05. (E) HepG2 and A549 cells were treated with miR-105 and miR-767 mimics or a negative control. Cell proliferation marker Ki-67 in (F) HepG2 and (G) A549 cells. Representative images show nuclear staining with Hoechst (blue) and Ki-67 staining (green). N = 3, *p < 0.05.

a bimodal module consisting of miR-1-1, miR-133a-1, miR-133a-2, and miR-133b. High expression was detrimental to overall survival in both cancer types (head and neck: HR: 1.41, CI: 1.08–1.85, p = 0.012; Stomach: HR: 1.39, CI: 1.01–1.91, p = 0.044; S5 Fig).

## Drug sensitivity in high miR-105 and miR767-expressing tumors

After showing that bimodal miRNA could be used to stratify patients based on overall survival, we then aimed to determine whether bimodal miRNA were also beneficial in predicting drug response. Instead of taking a comprehensive approach, we instead focused on the most common co-expression module, consisting of miR-105-1, miR-105-2 and miR-767. In head and neck, liver, lung, and stomach cancer, all three miRNA were expressed at very low levels in most tumors while a small number of patients had very high levels of expression (S6 Fig). Despite being located on the X chromosome, there was no difference in expression of miR-105-1, miR-105-2, and miR-767 between sexes (S7 Fig).

We found that high expression of miR-105-1, miR-105-2, and miR-767 resulted in earlier death among liver and lung cancer patients as well as higher cell proliferation *in vitro*. Therefore, tumors with high miR-105 and miR-767 expression appear to be particularly dangerous, and uncovering an effective treatment for this patient population is critical. In order to find effective drug treatments for high miR-105 and miR-767-expressing tumors, two steps were taken. First, we examined gene targets of miR-105 and miR-767. Secondly, we screened cancer drugs to see which were more effective on high miRNA-expressing cell lines. To understand the role of miR-105 and miR-767 in regulating gene expression, functional pathway analysis was performed on the top 1,000 predicted target genes of miR-105 and miR-767. It is important to note that miR-105-1 and miR-105-2 have the same seed sequence and thus the same putative targets. The PI3K-AKT signaling pathway was significantly enriched (false discovery rate p-value < 0.05; Fig 6A).

Confirmation of the gene targets within the PIK3-AKT pathway was performed by overexpressing miR-105 and miR-767 in HepG2 cells and measuring gene expression using qPCR (Fig 6B). Specifically, we measured genes that were predicted targets of miR-105 and miR-767 and/or play crucial roles in the PI3K-AKT signaling pathway. Out of the 44 genes tested, 12 were significantly downregulated after miRNA treatment, including Rho GTPase activating protein 35 (*ARHGAP35*), calcium/calmodulin dependent protein kinase 2 (*CAMKK2*), cyclin D2 (*CCND2*), cAMP responsive element binding protein 5 (*CREB5*), fibronectin 1 (*FN1*), forkhead box O3 (*FOXO3*), insulin receptor substrate 1 (*IRS1*), mitogen-activated protein kinase 1 (*MAPK1*), NRAS GTPase proto-oncogene (*NRAS*), PI3K catalytic subunit type 3 (*PIK3C3*), 3-phosphoinositide dependent protein kinase 1 (*PDK1*), and 6-phosphofructo-2-kinase/fructose-2,6-bisphosphatase 4 (*PFKFB4*). Two other genes, insulin-like growth factor 1 receptor (*IGF1R*) and KRAS GTPase proto-oncogene (*KRAS*), showed a trend for reduction by miRNA treatment (p < 0.1). Tyrosine 3-monooxygenase/tryptophan 5-monooxygenase activation protein epsilon (*YWHAE*) and PI3K catalytic subunit type 2 beta (*PIK3C2B*) were the only genes whose expression significantly increased with miRNA treatment.

Next, we investigated the relationship between miRNA levels and drug response. We first screened 13 cancer cell lines for miR-105 and miR-767 expression. To confirm the expression pattern of miR-105 and miR-767, we replicated results from patient samples by observing concurrent expression of miR-105 and miR-767 *in vitro* (S8 Fig). As a control, we also confirmed that miR-105 and miR-767 expression was not correlated with a random miRNA (miR-9; S8 Fig). We investigated drug response by pooling data from the Genomics of Drug Sensitivity in Cancer (GDSC) and Cancer Cell Line Encyclopedia (CCLE) databases. For each drug in each cell line, we downloaded the IC50 value, representing the drug concentration necessary to

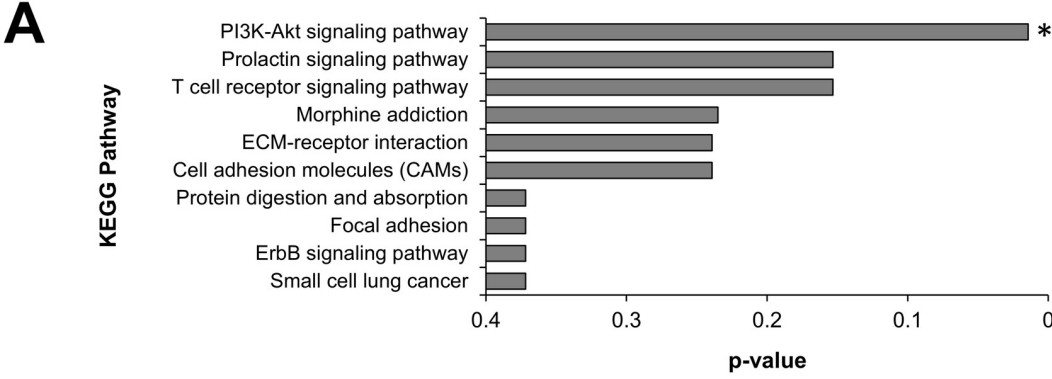

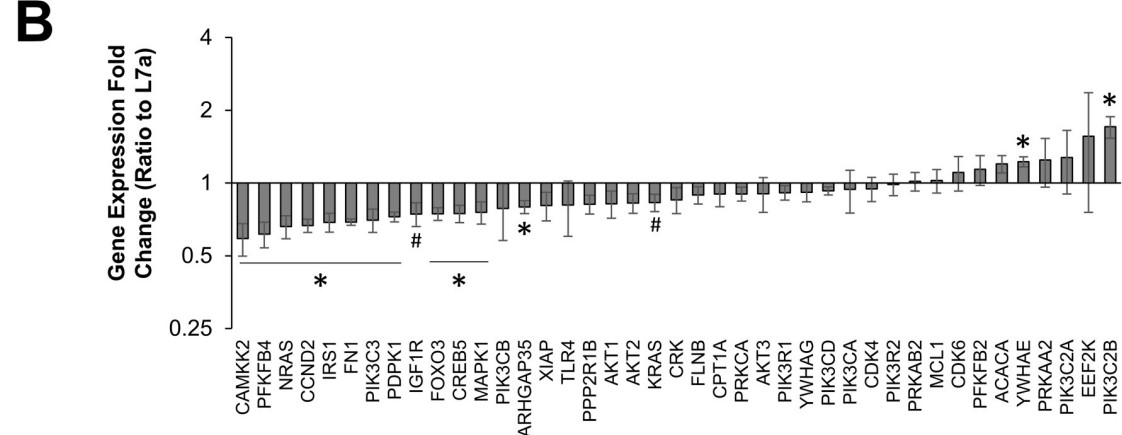

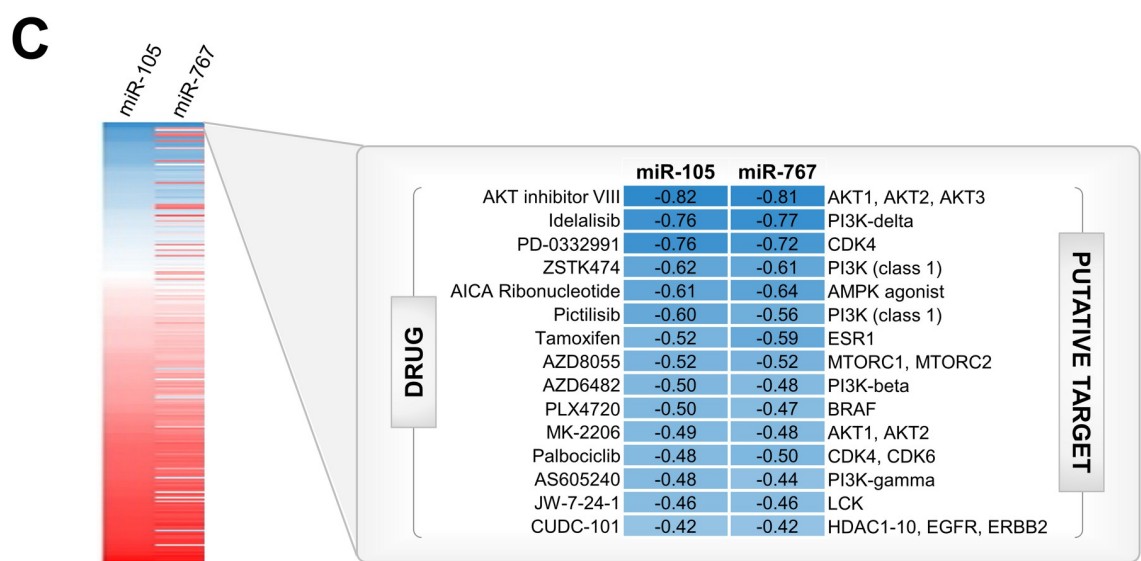

**Fig 6. miR-105 and miR-767 target genes in the PI3K-AKT pathway.** (A) KEGG pathway analysis was performed on the top 1,000 predicted gene targets of miR-105 and miR-767. False discovery rate corrected p-values are reported. (B) miR-105 and miR-767 were overexpressed in HepG2 cells. qPCR was used to measure expression of genes that were putative targets of both miR-105 and miR-767 and/or played a crucial role in the PI3K-AKT signaling pathway. $^*$p < 0.05. (C) The heat map shows correlations between miR-105/767 expression and drug IC50 values. Blue represents negative values while red denotes positive values. The top 15 drugs with the most negative correlation between miRNA and IC50 represent drugs that are most effective at killing high miR-105 and miR-767-expressing cell lines. Drug names are reported on the left and their putative protein targets are given on the right.

inhibit cell viability by 50%. We then computed Pearson correlations for each drug log(IC50) and miRNA expression (Fig 6C).

Low IC50 is indicative of greater drug potency; thus, drugs with a positive correlation between miRNA expression and IC50 (high miR-105/767 and high IC50) are less effective on high miR-105 and miR-767-expressing cells. Among the least effective drugs were two poly ADP-ribose polymerase (PARP) inhibitors (talazoparib and olaparib), the EGFR inhibitor gefitinib, and the sarcoplasmic/endoplasmic reticulum $Ca^{2+}$ ATPase (SERCA) inhibitor thapsigargin (S3 Table).

Conversely, drugs with a negative correlation between miRNA expression and IC50 (high miR-105/767 and low IC50) are effective in treating high miR-105 and miR-767-expressing cells. Out of the top 15 drugs which all had a correlation less than -0.4, eight directly targeted either PI3K (Idelalisib, ZSTK474, pictilisib, AZD6482, and AS605240), AKT (AKT inhibitor VIII and MK-2206), or AMPK (AICA ribonucleotide; Fig 6C). An additional three of the drugs targeted components of the PI3K-AKT and AMPK pathways, including cyclin dependent kinase (CDK4) (PD-0332991), CDK6 (palbociclib) and mammalian target of rapamycin complex 1 and 2 (mTORC1 and mTORC2) (AZD8055). Collectively, both the analysis of miRNA gene targets as well as the investigation of drug/miRNA correlations point to an interaction between miR-105/767 and the PI3K pathway. Therefore, we hypothesized that the PI3K pathway is a therapeutic target for high miR-105 and miR-767-expressing tumors.

Given the importance of the PI3K pathway, we next utilized *in vitro* assays to uncover whether high expression of miR-105 and miR-767 facilitated sensitivity to PI3K inhibiting drugs. We specifically used ZSTK474, a class I PI3K inhibitor that competitively binds the ATP binding site of PI3K. A549 cells were treated either with miR-105 and miR-767 mimics or with scrambled miRNA sequence as a control. After 24 hours of miRNA treatment, cells were treated with one of eight different concentrations of ZSTK474 and allowed to incubate for another 24 hours. Cell viability was measured using a WST-1 assay. When combined with the two miRNA, ZSTK474 more effectively reduced cell viability at low drug concentrations (0.16, 0.31, 0.63, and 1.25 μM; Fig 7A), but IC50 was not statistically lower in the miR-treated cells.

We then sought to determine the physiological processes mediating the decrease in cell viability. A549 cells were treated with concentrations of ZSTK474 ranging from 0 to 0.8 μM. After 8 hours of incubation, the cell cycle was measured using flow cytometry (Fig 7B). As ZSTK474 concentration increased, the percentage of cells in the S phase decreased while the percentage of cells undergoing G1 arrest increased (Fig 7C). Treatment with miR-105 and miR-767 mimics resulted in no difference in cell cycle progression compared to control (Fig 7D). Apoptosis was also quantified using five doses of ZSTK474 (Fig 7E). Measurements were taken at 4, 8, 12, and 16 hours after drug treatment. The total number of apoptotic cells was summed over all the time points and the fraction of cells in apoptosis was calculated (Fig 7F). Compared to control, miRNA treatment decreased the number of cells in early and late apoptosis when no ZSTK474 was present, but miRNA increased the number of cells in apoptosis at 0.1, 0.2, 0.4, and 0.8 μM ZSTK474 concentrations (Fig 7G and 7H). It is unclear why the percentage of cells in early apoptosis decreased at the highest drug concentration (0.8 μM). It is possible that this dose of ZSTK474 immediately induced apoptosis, and while we were able to capture late apoptosis, we missed the maximal early apoptotic effect.

We next investigated the molecular mechanisms underlying the miRNA-mediated increase in ZSTK474 sensitivity. First, PI3K and AKT gene expression were measured in response to miR-105 and miR-767 treatment alone, ZSTK474 alone, and the combination of miRNA and drug (ZSTK474+miR). miRNA treatment did not significantly inhibit any PI3K or AKT genes, but there was a trend for a reduction in *AKT1* expression (p = 0.056; S9 Fig). There was a

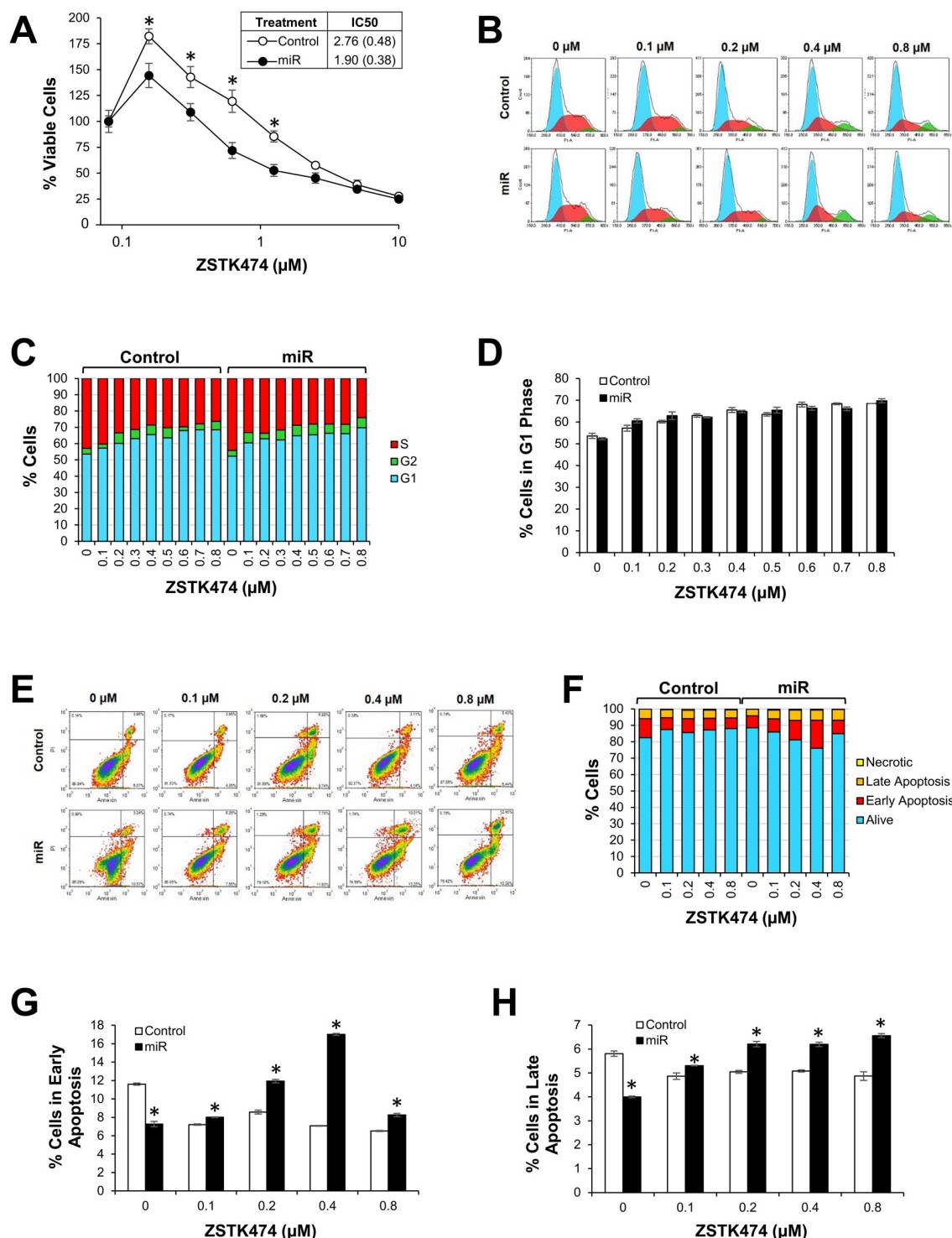

**Fig 7. miR-105 and miR-767 facilitate sensitivity to ZSTK474.** (A) ZSTK474 effects in A549 cells with miR-105 and miR-767 mimics or a negative control miRNA. Values are normalized to cells treated with 0 μM ZSTK474. IC50 values represent the ZSTK474 concentration that results in a half-maximal response. (B, C) Cell cycle analysis by flow cytometry. (D) The percentage of cells in the G1 phase. (E, F) Apoptosis was quantified using flow cytometry. Representative density plots show cells that are alive (lower left quadrant), in early apoptosis (lower right quadrant), in late apoptosis (upper right quadrant), and necrotic (upper left quadrant). miR-105 and miR-767 upregulate (G) early apoptosis and (H) late apoptosis upon ZSTK474 treatment. Data are presented as mean ± SEM. *p < 0.05.

significant effect of ZSTK474 on *PIK3CA*, *PIK3R1*, *PIK3R3*, and *AKT1* expression. Any change in gene expression was independent of miRNA treatment, suggesting that a reduction in PI3K and AKT expression were not responsible for the observed physiological differences between the ZSTK474 and ZSTK474 + miR conditions.

While major players of the PI3K pathway did not appear to account for the miRNA-induced increase in drug response, we hypothesized that proteins related to negative feedback and pathway crosstalk may impact drug sensitivity. We first performed a time-course experiment in which cells were incubated with miR-105 and miR-767 mimics for 24 hours and then treated with ZSTK474. Before drug treatment (0 hours) and at 0.5, 1, 2, and 4 hours after drug treatment, we measured proteins involved in negative feedback (IRS1 and FOXO3), proteins mediating PI3K signaling (PDK1), and proteins in the MAPK pathway (ERK and KRAS; Fig 8A). We also activated the PI3K pathway with insulin and measured protein levels after 2 hours of treatment (Fig 8B).

As expected, ZSTK474 treatment steadily reduced p-AKT levels over 4 hours (p = 6.3 x $10^{-4}$; Fig 8C). The converse was also true upon insulin treatment, as higher levels of p-AKT were observed (p = 2.01 x $10^{-9}$). Treatment with miRNA had no impact on p-AKT levels (p = 0.77). There was no change in total AKT levels with miRNA, ZSTK474, or insulin treatment (p = 0.25, p = 0.13. p = 0.37, respectively).

Activation of the PI3K pathway is controlled by negative feedback. mTOR and S6K can phosphorylate IRS1 and inactivate it [14–17]. Indeed, activation of the PI3K pathway with insulin treatment resulted in an increase in repressive phosphorylation of serine 302 on IRS1 (p = 8.9 x $10^{-5}$; Fig 8C). Conversely, ZSTK474-mediated downregulation of p-AKT was accompanied by loss of phosphorylated IRS1 (p-IRS1; p = 1.3 x $10^{-3}$). miRNA treatment did not impact p-IRS levels (p = 0.38), but did reduce total IRS1 protein expression (p = 1.4 x $10^{-2}$). Total IRS1 levels were also reduced by ZSTK474 (p = 1.8 x $10^{-3}$) and upregulated by insulin (p = 4.6 x $10^{-4}$). Negative feedback in the PI3K pathway is also achieved by FOXO3, a transcription factor that upregulates receptor tyrosine kinase (RTK) gene expression. AKT-mediated phosphorylation of FOXO3 (p-FOXO3) prevents translocation to the nucleus and consequently inhibits RTK expression [18]. Surprisingly, ZSTK474 treatment did not significantly reduce the levels of p-FOXO3 (p = 0.30). miR treatment had no effect on p-FOXO3 levels but a reduction in total FOXO3 was observed (p = 0.012).

Due to extensive crosstalk between the PI3K and MAPK pathways, we next measured MAPK pathway activation. ZSTK474 treatment immediately upregulated levels of phosphorylated ERK (p-ERK; p = 6.2 x $10^{-4}$; Fig 8C). This upregulation of p-ERK was partially attenuated by miR-105 and miR-767 (p = 4.4 x $10^{-2}$). We also measured PDK1 and KRAS, two proteins in the PI3K and MAPK pathways that were predicted targets of miR-105 and miR-767. While drug treatment reduced PDK1 levels (p = 3.5 x $10^{-2}$), miRNA treatment had no effect on protein levels. miRNA treatment also had no impact on KRAS expression.

Because protein levels of total FOXO3 and IRS1 were downregulated by miR-105 and miR-767, we examined whether the two miRNA mediated protein changes by directly binding the genes. FOXO3, IRS1, and PDK1 were predicted targets of both miR-105 and miR-767 (Fig 9A). We first measured mRNA expression and found decreased levels of *FOXO3*, *IRS1*, and *PDK1* after miRNA treatment (p = 2.8 x $10^{-2}$, p = 8.7 x $10^{-3}$, and p = 3.6 x $10^{-4}$, respectively; Fig 9B). Although ZSTK474 significantly reduced *PDK1* expression (p = 0.030), there was no effect of ZSTK474 treatment on *FOXO3* or *IRS1* expression. Thus, it appeared as though downregulation of *FOXO3*, *IRS1*, and *PDK1* was primarily miRNA-dependent rather than drug-dependent. In order to test whether *FOXO3*, *IRS1*, and *PDK1* were direct targets of miR-105 and/or miR-767, RNA immunoprecipitation was performed after miRNA treatment. When miRNA binds to mRNA, Argonaute 2 (AGO2) is recruited to catalyze endonuclease

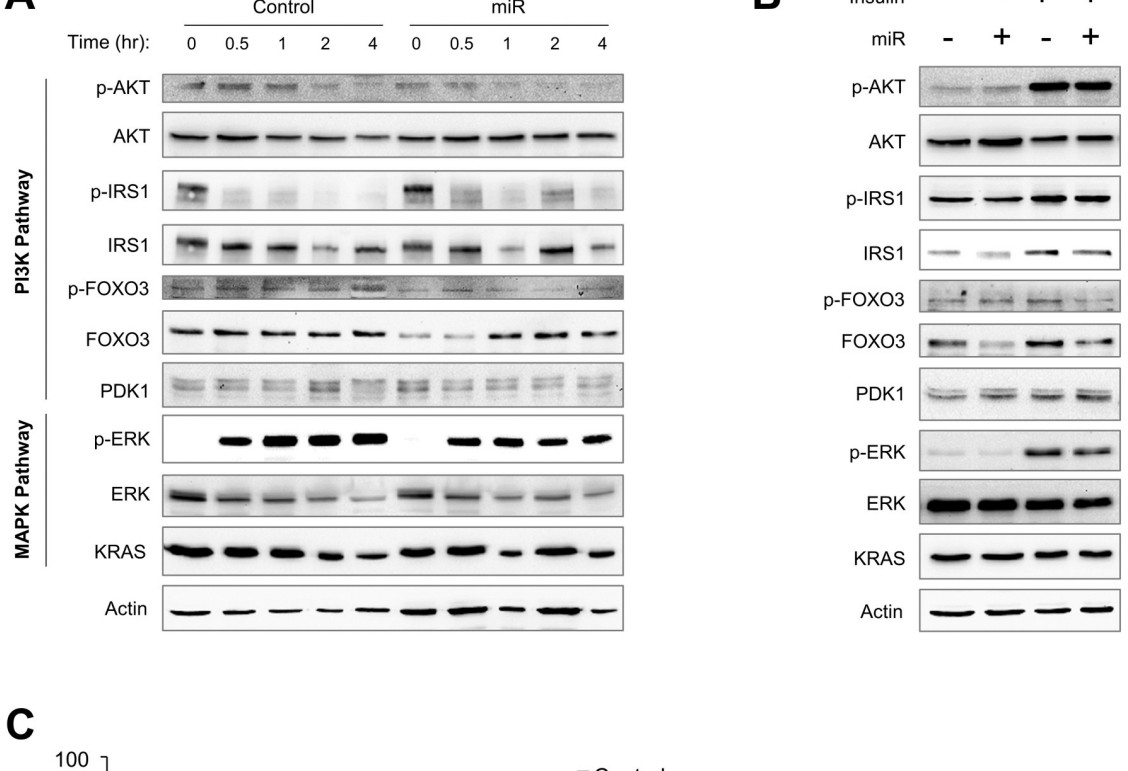

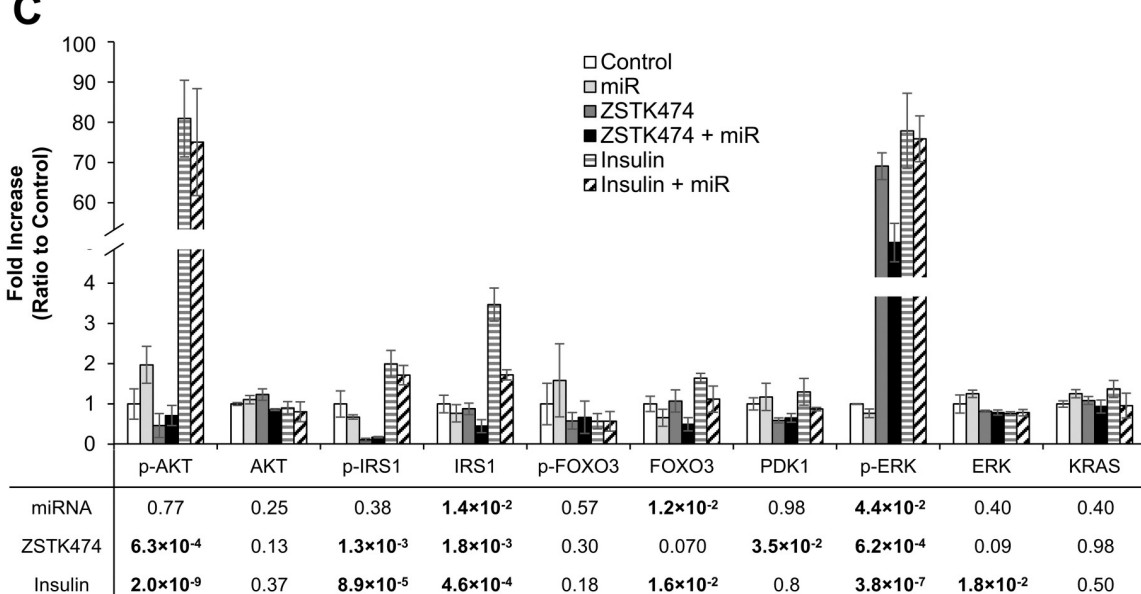

**Fig 8. Expression of PI3K and MAPK pathway proteins are mediated by miR-105 and miR-767.** (A) A549 cells were treated with negative control miRNA (Control) or miR-105 and miR-767 mimics (miR) for 24 hours, followed by 0.3 μM ZSTK474. Cells were collected at 0, 0.5, 1, 2, and 4 hours after drug treatment. (B) Control and miR-treated A549 cells were incubated with insulin for 2 hours and protein expression was measured. (C) Protein expression was quantified using three replicates per treatment condition. ANOVA was performed and the p-values of main effects of miRNA, ZSTK474, and insulin are reported. Data are normalized to Control and presented as mean ± SEM.

**A**

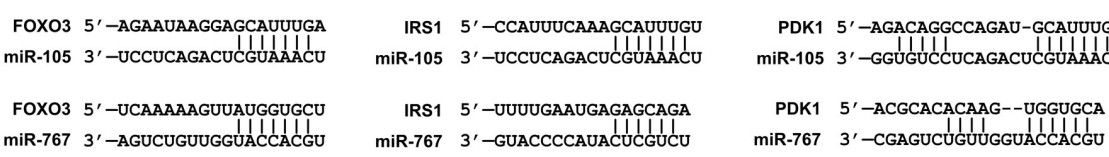

**B**

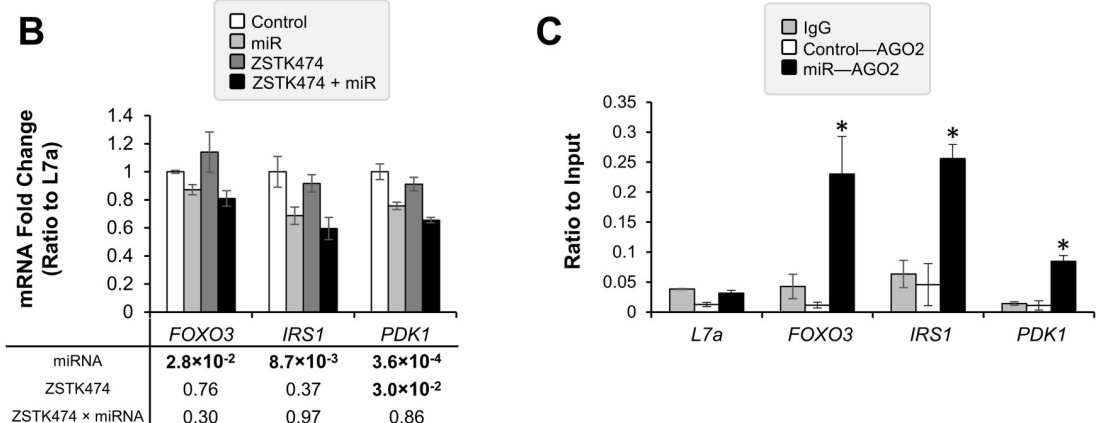

**C**

**D**

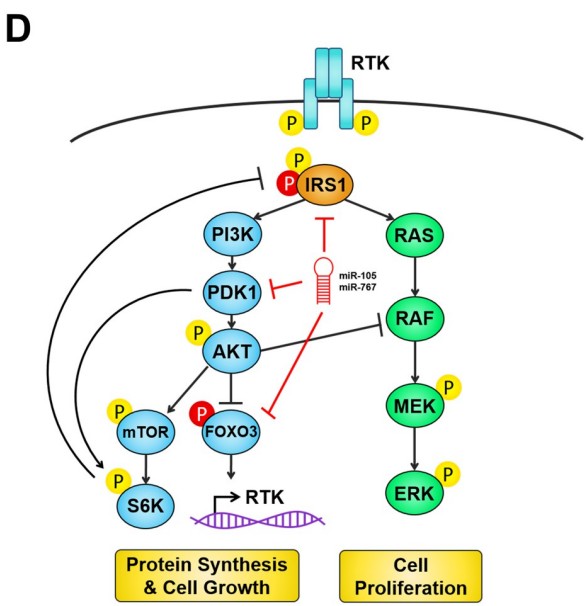

**Fig 9. miR-105 and miR-767 directly target genes in the PI3K pathway that mediate crosstalk with the MAPK pathway.** (A) Mature miR-105 and miR-767 sequences are predicted to bind complementary sequences in the 3'-UTRs of *FOXO3*, *IRS1*, and *PDK1*. (B) A549 cells were treated with miR-105 and miR-767 mimics and ZSTK474. Gene expression was measured using qPCR and normalized to L7a. Each treatment condition included three replicates. ANOVA was performed. The p-values for the main effect of miRNA and ZSTK474 as well as their interaction are reported. (C) RNA immunoprecipitation measured binding between AGO2 protein and *FOXO3*, *IRS1*, *PDK1 mRNA*. Data are presented as mean ± SEM. *Denotes a significant difference between Control and miR (p < 0.05). (D) Crosstalk between the PI3K and the MAPK pathways arises from AKT inhibition of RAF, IRS1 activation of RAS, and FOXO3-mediated transcription of RTKs. miR-105 and miR-767 overexpression resulted in direct binding to IRS1, PDK1, and

FOXO3, and decreased levels of phosphorylated ERK. Blue: PI3K pathway proteins, green: MAPK pathway proteins, yellow P: activating phosphorylation, red P: inactivating phosphorylation.

cleavage. Higher levels of *FOXO3*, *IRS1*, and *PDK1* mRNA were bound to AGO2 when cells were treated with miR-105 and miR-767, suggesting that the genes are direct targets of miR-105 and/or miR-767 (Fig 9C). Overall, based on results presented in this manuscript, we propose that miR-105 and miR-767 target genes that inhibit undesirable drug effects. Specifically, the miRNAs target FOXO3 and IRS1 to diminish drug-induced over-activation of the MAPK pathway (Fig 9D).

## Discussion

In the present study, a novel methodology was applied to identify bimodal miRNA across cancer types. To our knowledge, this is the first study to analyze the functional role of bimodal miRNA. Furthermore, we are the first to investigate large-scale bimodal expression patterns across cancers using next-generation sequencing data from clinical samples. We showed that high levels of bimodal miRNA expression was characteristic of cancer. Furthermore, several bimodal miRNA were common to multiple cancer types, suggesting that certain miRNA consistently account for tumor heterogeneity and may be involved in general oncogenic processes. The relevance of these bimodal miRNA was confirmed by showing that they could be used to predict overall survival and drug response. To illustrate the importance of bimodal miRNA, we specifically focused on miR-105 and miR-767. The two miRNA were bimodally expressed in liver and lung cancer, and high expression was indicative of poor prognosis. Furthermore, high miR-105 and miR-767-expressing cells responded better to PI3K inhibiting drugs. We demonstrate that bimodal miRNA are viable biomarkers in cancer and may equip physicians to better understand the patient prognosis and devise effective treatment strategies.

We utilized our controlled mixture modeling approach to identify bimodal miRNA that played a role in cancer. The inclusion of control samples appeared to reduce false positives. We assumed that if genes followed a similar distribution in the cancer population and the general population, then they were not relevant to tumorigenesis. Not only did CM rank known bimodal breast cancer genes more favorably, but it also excluded genes with known sequence variation (immunoglobulins) and high copy number variation (glutathione transferase) in the general population [19]. We are the first to implement control tissue as a means of addressing this issue. Previous studies have simply removed known bimodal genes from the analysis [19], but our method still considers those genes while imposing a penalty. Thus, we account for the possibility that known bimodal genes may contribute to tumorigenesis, but only if they present a different distribution than controls. Furthermore, our method accounts for genes that have not yet been discovered as bimodal in the general population. This is particularly important for the analysis of bimodal miRNA. Unlike protein-coding genes whose population sequence and copy number variation have been extensively studied, there is limited information regarding miRNA expression distributions in the healthy population. Therefore, removing miRNA based on prior knowledge would be impossible.

We specifically used our methodology to identify bimodal miRNA across cancers. First, we found that kidney cancer had the most bimodal miRNA, which may be indicative of greater heterogeneity. This may be due to the inclusion of two subtypes, including clear cell and papillary cell carcinoma. Previous research has shown that kidney tumor subtypes differ in their gene expression, mutations, non-coding RNAs, and DNA methylation, leading to a heterogeneous molecular landscape [20]. We also found that thyroid cancer had bimodal miRNA that

were highly related to each other, suggesting two subtypes of patients with high genetic concordance within each group. The two most common types of thyroid cancer, papillary and follicular carcinoma, been defined by different somatic mutations [21], gene expression [22–24], and miRNA profiles [25]. Therefore, the large number of bimodal miRNA that we observed may be reflective of the two subgroups. In our analysis thyroid cancer had the largest concurrently-expressed miRNA module, containing 14 miRNA. This is supported by prior studies which have shown consistent molecular changes specifically within papillary or follicular thyroid carcinomas [26].

As opposed to kidney and thyroid cancer, breast and prostate cancer had fewer bimodal miRNA, suggesting that there is more genetic similarity amongst tumors than in other types of cancer. Both cancers were represented exclusively by one sex, which may have resulted in less transcriptional variation. However uterine cancer had the second highest number of miRNA with bimodality index thresholds > 1.4 or 1.5, so low variation due to lack of sex differences does not totally account for the small number of bimodal miRNA in breast and prostate cancer. Unlike uterine cancer which was divided into endometrial carcinoma (TCGA-UCEC), carcinosarcoma (TCGA-UCS), and sarcoma (TCGA-SARC) subtypes, prostate, and breast cancer were each classified under one group in GDC (TCGA-BRCA and TCGA-PRAD), and both primarily impact glandular cells. Prior studies have demonstrated large expression differences between cell types and high concordance within cell types, so having one predominantly affected cell type in prostate and breast cancer may reduce variation [27, 28]. For breast and prostate cancer, several classifications based on miRNA and gene expression have been devised [29–35], but it is unclear how variability in miRNA expression compares to other cancer types. We provide evidence that miRNA expression is more normally distributed in breast and prostate cancer, which may pose a greater challenge for miRNA-based patient stratification in these populations.

We also compared between cancers and found that certain bimodal miRNA modules were common to multiple cancer types, which may suggest genetic similarities between certain cancer types. For instance, head and neck cancer had two bimodal modules in common with lung cancer. Interestingly, both cancers consisted of squamous cell carcinoma cases, which have been shown to have the similar gene and miRNA expression patterns [28, 36, 37], SNPs, and risk factors, such as smoking and high fasting glucose levels [38]. Thus, the two cancers may be impacted by the same carcinogens and oncogenic driver genes, resulting in similar miRNA profiles. Furthermore, head and neck and lung cancers also shared one bimodal module with uterine cancer. Fewer similarities have been found between uterine cancer and the other two, but basal and secretory lung cancer, mesenchymal head and neck cancer, and immunoreactive uterine cancer have been shown to display similar gene expression patterns and greater immune activation [36]. Finally, head and neck, liver, lung, and stomach cancer shared a bimodal expression of miR-105 and miR-767. Head and neck, liver, and lung cancer have been shown to have similar expression patterns in cell cycle genes [37], but the previous investigation has not identified many genetic similarities between stomach cancer and the other three cancer types. Gastrointestinal cancers tend to have similar miRNA profiles [28], however, lung adenocarcinoma and stomach adenocarcinoma also display similar miRNA patterns [39]. Therefore, overlap in bimodal miRNA expression between cancer types could suggest that certain cell types are likely to display similar genetic profiles.

Alternatively, consistent bimodal miRNA expression between cancers may indicate the importance of such miRNA in general oncogenic processes. Hallmarks of cancer include such characteristics as tissue invasion and metastasis, replicative immortality, evasion of apoptosis, and metabolic reprogramming [1]. However, distinct aberrant phenotypes may arise within the hallmarks. Indeed, the progression from a mild tumorigenic state to another more severe

phenotype has been shown to be associated with genetic changes. For instance, invasion and metastasis are regulated by EMT, which marks the transition from a polar epithelial cell to a mobile mesenchymal cell. Genetic regulation of EMT involves transcription factor activation to induce cytoskeletal reorganization, including loss of intercellular adhesion proteins, such as E-cadherin and occludins, and gain of vimentin and smooth muscle actin [40]. Also, epigenetic factors like DNA hypomethylation of EMT transcription factors [41–43] and hypermethylation of E-cadherin are associated with EMT [44–46]. Another hallmark of cancer is reprogrammed cellular energetics. While aerobic glycolysis-dependent tumors represent one aberrant metabolic phenotype, a different hybrid phenotype utilizes both glycolysis and oxidative phosphorylation. Mitochondrial signaling to the nucleus as well as oncogenes such as RAS, MYC, and c-SRC have been shown to mediate the relationship between glycolysis and oxidative phosphorylation [47, 48]. Thus, it may be the case that bimodal miRNA are associated with bimodal physiological states. Because many of the identified bimodal miRNA were common to several cancer types, we hypothesize that such oncogenic states are not cancer-specific, but rather are broadly observed across cancers. It is unclear whether miRNA expression might be driving or responding to physiological differences. Future studies should determine concrete associations between bimodal miRNA and universal oncogenic features.

In order to show the clinical utility of bimodal miRNA, we first examined whether they could be used to predict overall survival. We found that multiple bimodal miRNA modules could successfully predict overall survival in five out of eight cancer types. We also examined single miRNA modules. High miR-105 and miR-767 expression were indicative of poor survival in liver and lung cancer. This is consistent with past reports suggesting that miR-105 and miR-767 may play an oncogenic role in melanoma, breast, lung, and colorectal cancer. In colorectal and breast cancer, miR-105 was upregulated in tumor compared to control tissue, and high expression was indicative of poor survival outcomes [49–51]. miR-105 overexpression was also shown to induce NF-κB signaling [52] and heighten EMT in colon cancer cell lines and mouse tumors [53]. Similarly, miR-767 was shown to promote cellular proliferation in human melanoma cell lines [54] while miR-105 overexpression increased cell viability in non-small cell lung cancer cell lines [55]. Additionally, miR-105 and miR-767 were shown to target ten-eleven translocation tumor suppressor genes (TET1 and TET3), implicating miR-105 and miR-767 as putative oncogenes [56]. Finally, exosomally secreted miR-105 was shown to promote breast cancer metastasis by reprogramming neighboring fibroblasts and breaking endothelial tight junctions [57, 58].

In contrast, several studies point to a tumor suppressive role for miR-105 and miR-767. Specifically, high miR-767 was associated with longer overall survival in thyroid cancer patients [59], while high miR-105 expression was correlated with more favorable survival in non-small cell lung cancer, glioma, and hepatocellular carcinoma patients [60–64]. Furthermore, in both glioma and prostate cancer, miR-105 overexpression *in vitro* reduced cellular proliferation and induced G1 arrest, while inhibiting tumor growth *in vivo* [65, 66]. Collectively, data suggests that the effects of miR-105 and miR-767 in tumorigenesis are variable between cancer types and should be investigated in future experimentation.

In addition to overall survival, we also examined the ability of bimodal miRNA to predict drug response. We found that miR-105 and miR-767 overexpression potentiated the PI3K inhibitor-mediated decrease in cell viability. High miR-105 and miR-767 levels created a cellular environment that complemented the action of PI3K inhibitors by mitigating off-target effects. PI3K inhibitors may have undesirable effects that arise from negative feedback and crosstalk between related pathways. Blocking AKT signaling has beneficial actions, including reduced activation of mTOR [67] and NF-κB [68] to decrease protein synthesis and increase apoptosis. However, it also reduces negative feedback on the pathway. For instance, AKT

typically phosphorylates FOXO3 and prevents it from entering the nucleus and acting as a transcription factor for RTKs [69, 70]. Inhibition of PI3K can also decrease negative feedback by S6K on IRS1 [14, 15]. Both of these mechanisms impact not only the PI3K pathway itself but also affect the MAPK pathway, which is activated by the same RTKs and IRS1. We demonstrated this crosstalk by showing a sharp induction of p-ERK after PI3K inhibition. Although the precise mechanism necessary for this increase is unclear, we show that the increase is partially mitigated by miR-105 and miR-767 and hypothesize that this is the consequence of reduced levels of FOXO3 and IRS1 (Fig 9D). We demonstrate one possible avenue by which bimodal miRNA may impact drug sensitivity, but additional investigation might utilize Argonaute high-throughput sequencing of RNA isolated by crosslinking immunoprecipitation (AGO HITS-CLIP) to identify all miRNA-mRNA interactions [71, 72].

While we are the first to suggest a role for miR-767 in PI3K signaling, previous reports have shown that miR-105 directly interacts with proteins in the PI3K pathway. In liver cancer, miR-105 has been hypothesized to play a tumor suppressing role via its interaction with genes in the PI3K pathway [73]. We confirmed downregulation of IRS1 protein and both *IRS1* and *PDK1* mRNA. We also found an increase in AGO2 binding to *IRS1* and *PDK1* with miRNA treatment. However, we did not find any difference in total AKT or PDK1 protein. Also, Shen et al. found lower levels of p-FOXO3 but report no changes in FOXO3 protein with miR-105 treatment. We found decreased FOXO3 protein levels. It is unclear exactly why these discrepancies exist, but we hypothesize that concurrent treatment with miR-767 may account for the inhibition of some protein targets as well as differences in cell proliferation. Also, our study used A549 lung cancer cells for drug and miRNA studies, while Shen et al. used HepG2. Due to inconsistent genetic backgrounds and baseline gene expression, it is possible that miRNA-induced changes in protein expression and cell survival could be more easily observed in certain cell populations. Thus, it may be important to test multiple cell lines in order to determine whether miR-105 and miR-767 have a universal mechanism of action or whether they perform specific functions in different cell types. Overall, our findings confirm that miR-105 and miR-767 are important for PI3K signaling, and we add to the existing body of literature by demonstrating their ability to identify which drugs may be effective in treating particular subgroups of tumors.

The current study has several strengths and limitations. Previous bimodality studies have focused exclusively on protein-coding genes, but our investigation broadens this scope, as we are the first to apply our novel method and examine bimodal miRNA. Although our method is straightforward and can be easily implemented in a clinical setting, there are limitations to the simplicity of our approach. Tumors are genetically complex, and modules of bimodal miRNA are ultimately part of a larger tumor profile. Incorporating networks of miRNA, genes, and other microenvironment parameters may be necessary to better predict patient outcome. Our *in vitro* analysis of miR-105 and miR-767 may also benefit from more omics data and statistical learning methods to fully understand the underlying biological state in order to identify the primary drivers of tumorigenesis.

Additional investigation should also aim to uncover the biological mechanisms by which miR-105 and miR-767 mediate drug sensitivity. We utilized computational tools to identify gene targets of miR-105 and miR-767 in the PI3K pathway. *In vitro* results also showed downregulation of IRS1 and FOXO3 with miR treatment, but without the appropriate knockdown experiments, we cannot conclude that these gene targets are directly responsible for increasing drug sensitivity. Since miR-105 and miR-767 have targets outside the PI3K pathway, it is possible that other genes may play a role. Future experiments should utilize Argonaute high-throughput sequencing of RNA isolated by crosslinking immunoprecipitation (AGO HITS-CLIP) to identify all miRNA-mRNA interactions [71]. Finally, other factors such as baseline

gene expression and tumor microenvironment may mediate miRNA effects, so incorporating additional cell lines as well as in vivo studies will be crucial in finding the link between miR-105 and miR-767 and drug response.

Although we only examined single miRNA modules, we provide preliminary evidence that bimodal miRNA may impact the clinical outcome, and we devise a foundational methodology that can be expanded to yield a more comprehensive patient profile.

## Conclusion

In the current study, we identified bimodal miRNA across cancer types and showed how they could be used for patient stratification based on prognosis and drug response in several types of cancer. We first devised an approach for identifying bimodal miRNA and applied our model to miRNA-seq data. We are the first to examine genome-wide bimodal expression across cancers using sequencing data. We found that certain miRNA were bimodally expressed in multiple cancer types, suggesting that they may be associated with general oncogenic characteristics. Specifically, we examined the importance of miR-105 and miR-767 in predicting overall survival in liver and lung cancer as well as facilitating sensitivity to PI3K inhibiting drugs. Our study provides a framework for finding bimodal expression and demonstrates the role of bimodal miRNA in tumorigenesis as well as their potential in predicting patient survival and enabling effective treatment.

## Methods

### Genomic datasets

RNA-seq data for methodological validation was downloaded from Genomic Data Commons (GDC, formerly TCGA). We decided to test our model on breast cancer mRNA expression data (Project ID: TCGA-BRCA), given the large number of patient tumor (n = 1,102) and control samples (n = 113), as well as the extensive characterization of specific bimodal genes in previous literature (e.g. estrogen receptor 1 (*ESR1*), human epidermal growth factor receptor 2 *(HER2 or ERBB2)*, progesterone receptor (*PGR*), etc.) [74, 75]. RNA-seq data were normalized according to gene length and total number of reads mapped, such that values were expressed as reads per kilobase of transcript per million mapped reads (RPKM).

For identifying novel tumorigenic miRNA, miRNA-seq data was downloaded from GDC for nine types of tumors: breast (Project ID: TCGA-BRCA; n = 1,096), head and neck (H&N; Project ID: TCGA-HNSC; n = 523), kidney (Project ID: TCGA-KIRC, TCGA-KIRP; n = 835), liver (Project ID: TCGA-LIHC; n = 372), lung (Project ID: TCGA-LUAD, TCGA-LUSC; n = 997), prostate (Project ID: TCGA-PRAD; n = 498), stomach (Project ID: TCGA-STAD; n = 446), thyroid (Project ID: TCGA-THCA; n = 506), and uterine cancer (Project ID: TCGA-UCEC, TCGA-UCS, TCGA-SARC; n = 861). Additionally, miRNA-seq data was downloaded from GDC for normal tissue, including breast (n = 104), head and neck (n = 44), kidney (n = 105), liver (n = 50), lung (n = 91), prostate (n = 52), stomach (n = 45), thyroid (n = 59), and uterine (n = 30). These nine cancers were chosen due to data availability. Tumor samples were pooled from all stages and both genders.

### Controlled mixture modeling (CM) and bimodality index calculation

Mixture modeling was performed using the same method for both the methodology validation on mRNA and the novel discovery of tumorigenic miRNA. Initially, genomic data were processed such that all data were log2 transformed and very lowly expressed miRNA were excluded from the analysis. Each miRNA was analyzed individually in each cancer type.

Our controlled mixture modeling (CM) incorporated control samples to assess bimodality of miRNA using the following steps:

1. Within each tumor type, every miRNA was fit with a one- and two-component Gaussian mixture model. The one-component model followed a Gaussian distribution with mean μ and variance σ:

$$f_1(x_i|\theta) = G(\mu, \sigma),$$

The two-component model was a mixture of two Gaussians, represented by the density function

$$f_2(x_i|\theta) = \pi_{1T}G(\mu_{1T}, \sigma_{1T}) + \pi_{2T}G(\mu_{2T}, \sigma_{2T}).$$

For each gene, $x_i$, in the tumor samples, $\mu_{1T}$ and $\mu_{2T}$ are the means of each component such that $\mu_{1T} < \mu_{2T}$. Also, $\pi_{1T}$ and $\pi_{2T}$ represent the proportion of observations in each component. To avoid miRNA with a small number of outliers, we required $\pi_{1T}$ and $\pi_{2T}$ to both be greater than 0.1.

2. The expectation maximization (EM) algorithm followed by computation of the Bayesian information criterion (BIC) was used to determine whether one- or two-component model was a better fit for the data. All miRNA better fit by one component were disregarded as unimodal. miRNA better fit by two components were re-clustered using k-means with k = 2.

3. For two-component miRNA, one- and two-component Gaussian mixture models were fit to the same miRNA from control tissue samples. The EM algorithm plus BIC was used to decide whether one or two components better fit the data. This resulted in one of two possibilities:

    1. The miRNA was unimodally distributed amongst control samples. In this case the bimodality index of the miRNA was calculated by

$$BI = \sqrt{\pi_{1T} \times \pi_{2T}} \frac{|\mu_{1T} - \mu_{2T}|}{\sqrt{\pi_{2T}\sigma_{1T}^2 + \pi_{1T}\sigma_{2T}^2}} \ .$$

    2. The miRNA was bimodally distributed amongst control samples. In this case, the miRNA was better fit by the two-component model and could be represented by the density function

$$f_2(x_i|\theta) = \pi_{1C}G(\mu_{1C}, \sigma) + \pi_{2C}G(\mu_{2C}, \sigma).$$

    Similarly to the model used for tumor samples, $\mu_{1C}$ and $\mu_{2C}$ are the means of each component in the control samples such that $\mu_{1C} < \mu_{2C}$, and $\pi_{1C}$ and $\pi_{2C}$ represent the proportion of observations in each component. Additionally, only $\pi_{1C}$ and $\pi_{2C}$ values greater than 0.1 were considered.
    K-means with k = 2 was then used to re-cluster the miRNA in control samples. The bimodality index was penalized and calculated in two ways

$$BI_{\mu \ penalty} = BI - \left( \frac{1}{|\mu_{1C} - \mu_{1T}| + |\mu_{2C} - \mu_{2T}|} \right)$$

as well as

$$BI_{\pi\ penalty} = BI - \left( \frac{1}{|\pi_{1C} - \pi_{1T}| + |\pi_{2C} - \pi_{2T}|} \right).$$

$BI_{\mu\ penalty}$ and $BI_{\pi\ penalty}$ were compared and the larger value was chosen as the final bimodality index for the miRNA.

By penalizing the bimodality index calculation, we attempt to filter false positives when a miRNA is bimodally expressed in the control population. The $BI_{\mu\ penalty}$ greatly penalizes the bimodality index score when the means of the components are similar between tumor and control. The $BI_{\pi\ penalty}$ penalizes the bimodality index score when the components are similarly partitioned between tumor and control. We compared our method to an unpenalized bimodality index (i.e. mixture model plus k-means [MM] without control samples).

## Survival analysis

Survival analysis was performed to determine differences in overall survival time. First, data were fit using a Cox proportional hazards regression. Next, hazard ratio (HR) was calculated and a Wald test was performed to test for significance. Median survival time and 95% confidence intervals were calculated and reported with Kaplan Meier curves.

## Target prediction and pathway analysis

Gene targets of miR-105 and miR-767 were downloaded from TargetScan version 7.2 [76]. The top 1,000 predicted genes were used for functional pathway analysis in DAVID version 6.8 [77]. KEGG pathways with false discovery rate p-value $< 0.05$ were considered statistically significant.

## Drug sensitivity analysis

To identify drugs that effectively inhibited growth in high miR-105 and miR-767-expressing cells, we looked at correlations between miR-105/767 and IC50 values for different drugs. We used the same cell lines in which we quantified miRNA expression (HCT116, HepG2, HT29, Jurkat, KM, MB231, MB435, MCF7, RKO, SKOV3, SW480, SW620, and T47D). For each drug in each cell line, we downloaded the IC50 values for drugs from the Cancer Cell Line Encyclopedia (CCLE) and Genomics of Drug Sensitivity in Cancer (GDSC) databases [78, 79]. In total, 14 drugs from CCLE and 214 drugs from GDSC were examined. We then computed Pearson correlations for each drug log(IC50) and miRNA expression.

## Cell lines and treatments

Thirteen cell lines were utilized for miRNA quantification. For drug sensitivity studies, A549 and HepG2 cells were used. Cells were cultured according to a standard procedure in our laboratory [80]. miRNA overexpression was performed using miRNA mimics with a scrambled miRNA sequence used as a control. For all experiments using ZSTK474 treatment, cells were first treated with either miR-105 and miR-767 mimics or negative control for 24 hours before ZSTK474 treatment was administered. The transfection protocol is outlined in the S1 Supplementary Methods.

### miRNA and mRNA expression

Gene expression was measured using quantitative real-time PCR (qPCR). All mRNA primers are described in S4 Table. Gene expression was normalized to the expression of the internal control gene ribosomal protein L7a (*L7a*). For miRNA expression, qPCR was performed using TaqMan miRNA Assay primers that were specific to miR-105 or miR-767 (Applied Biosystems) and miRNA were quantified via the ΔΔCT method. A TaqMan miRNA Control Assay for U6 snRNA was included for normalization. miRNA primers can be found in S5 Table. A detailed PCR protocol can be found in the S1 Supplementary Methods.

### Immunofluorescence and cell viability assay

Cells grown on coverslips and incubated in Ki-67 primary antibody (BioLegend, San Diego, CA; S6 Table) then incubated with Hoechst 33342 (Invitrogen; S6 Table). Cells were imaged via light microscopy at magnification of 63X. To quantify expression, three 67.74 μm x 67.74 μm area were randomly selected in each group, and analysis was performed using ImageJ software. Cell viability was determined by WST-1 assay. A detailed immunofluorescence and cell viability assay procedure can be found in the S1 Supplementary Methods.

### Flow cytometry

Flow cytometry was used for cell cycle and apoptosis analysis. Cells were incubated in fluorescein-conjugated annexin V (FITC Annexin V, BioLegend, San Diego, CA) and propidium iodide (PI, Sigma-Aldrich, St. Louis, MO). Unstained cells were considered non-apoptotic, annexin V positive/PI negative cells were considered early apoptotic, and annexin V positive/PI positive cells were considered late apoptotic. Specific cell preparation and flow cytometry settings are outlined in the S1 Supplementary Methods.

### Western blot analysis

In order to assess the effects of miR-105 and miR-767 on protein levels, the two miRNA were overexpressed *in vitro* and western blot was used to quantify proteins in the PI3K pathway. For quantification, samples were run in triplicate. Total protein was normalized to beta actin and phosphorylated proteins were normalized to the corresponding total protein. A detailed western blot protocol can be found in the S1 Supplementary Methods. All antibodies can be found in S6 Table.

### RNA immunoprecipitation

RNA immunoprecipitation (RIP) was used to quantify binding between miRNA and putative target mRNA. High complementarity between the miRNA and mRNA catalyzes endonuclease cleavage via AGO2. For each RIP, A549 cells were incubated with miR-105 and miR-767 or control miRNA. Antibodies for AGO2 and IgG (control) were utilized. RNA was then purified and total RNA quantification, cDNA synthesis, and qPCR were performed as described above. RIP methodology is further detailed in the S1 Supplementary Methods.

### Additional statistical analysis

All *in vitro* experiments were performed in triplicate. Pairwise comparisons were made using student's t-tests. For drug and miRNA experiments involving multiple treatment conditions, ANOVA was performed. Data are presented as mean ± SEM. P-values < 0.05 were considered statistically significant.

## Supporting information

**S1 Table. Concurrently expressed bimodal miRNA modules.** Pairwise Pearson correlations were calculated between miRNA with bimodality index > 1.4. Reported modules contain at least three miRNA with correlations > 0.5 between all miRNA pairs.
(PDF)

**S2 Table. Survival analysis using multiple bimodal miRNA modules.** Hierarchical clustering was performed using bimodal miRNA co-expression modules, and patients were divided into two groups. Using the two groups, a Cox proportional hazards regression was fit for each cancer type. The reported hazard ratio (HR) denotes the risk of death in Group 2 compared to Group 1. The statistical significance (p-value) of each HR was determined using a Wald test. MST: median survival time, CI: confidence interval.
(PDF)

**S3 Table. Drug sensitivity based on miR-105 and miR-767 expression.** Values represent correlations between miRNA expression and drug log(IC50) value.
(PDF)

**S4 Table. mRNA primers.** qPCR amplification efficiency is calculated based on the slope of the standard curve. Slopes between -3.30 ± 0.20 and amplification efficiencies of 100 ± 10% are typically considered acceptable.
(PDF)

**S5 Table. miRNA probes.** All primers are Taqman Advanced miRNA Assay.
(PDF)

**S6 Table. Antibody information.** IF: immunofluorescence, WB: western blot, RIP: RNA immunoprecipitation.
(PDF)

**S1 Fig. Identification of bimodal miRNA using mixture modeling with k-means.** Bimodally expressed miRNA were examined in nine types of cancer and control tissue using mixture modeling with k-means (MK). Analysis was performed using bootstrapped samples of the same size as control tissue. Graphs show the number of bimodal miRNA at bimodality index (BI) thresholds: (A) BI > 1.2, (B) BI > 1.3, (C) BI > 1.4, (D) BI > 1.5.
(TIFF)

**S2 Fig. Survival analysis using miR-105 and miR-767.** (A) Head and neck and (B) stomach cancer patients were divided into low and high miRNA expressers (blue and red, respectively). For each cancer type, a Cox proportional hazards regression was fit. The reported hazard ratio (HR) and 95% confidence interval (CI) denote the risk of death in the high expressing group compared to the low expressing group. The statistical significance (p-value) of each HR was determined using a Wald test. Median survival time is given in days with a 95% CI. (C) The relationship between miRNA expression and TNM stage was examined. Graphs show the proportion of individuals with high (red) and low (blue) miR-105 and miR-767 expression within each stage. Differences in TNM staging between high and low expressers was tested using Chi square tests, but there were no significant findings.
(TIFF)

**S3 Fig. Survival analysis using miR-9.** (A) miR-9-1, miR-9-2, and miR-9-3 are located on chromosome 1, 5, and 15, respectively. (B) Head and neck, (C) lung, and (D) uterine cancer patients were divided into low and miRNA expressers (blue and red, respectively). For each

cancer type, a Cox proportional hazards regression was fit. The reported hazard ratio (HR) and 95% confidence interval (CI) denote the risk of death in the high expressing group compared to the low expressing group. The statistical significance (pvalue) of each HR was determined using a Wald test. Median survival time is given in days with a 95% CI. (E) The relationship between miRNA expression and TNM stage was examined. Graphs show the proportion of individuals with high (red) and low (blue) miR-9 expression within each stage. Differences in TNM staging between high and low expressers was tested using Chi square tests. $^*$p<0.05.
(TIFF)

**S4 Fig. Survival analysis using miR-96, miR-182, and miR-183.** (A) miR-96, miR-182, and miR-183 are located in an intergenic region of chromosome 7. (B) Kidney and (C) liver cancer patients were divided into low and miRNA expressers (blue and red, respectively). For each cancer type, a Cox proportional hazards regression was fit. The reported hazard ratio (HR) and 95% confidence interval (CI) denote the risk of death in the high expressing group compared to the low expressing group. The statistical significance (p-value) of each HR was determined using a Wald test. Median survival time is given in days with a 95% CI. (D) The relationship between miRNA expression and TNM stage was examined. Graphs show the proportion of individuals with high (red) and low (blue) miR-96, miR-182, and miR-183 expression within each stage. Differences in TNM staging between high and low expressers was tested using Chi square tests. $^*$p<0.05.
(TIFF)

**S5 Fig. Survival analysis using miR-1, miR-133a, and miR-133b.** (A) miR-1-1 and miR-133a-2 are located in an intronic region on chromosome 20. miR-133a-1 is located within the miR-133a-1 host gene (MIR133A-1HG). miR-133b is located in an intergenic region of chromosome 6. (B) Head and neck and (C) stomach cancer patients were divided into low and high miRNA expressers (blue and red, respectively). For each cancer type, a Cox proportional hazards regression was fit. The reported hazard ratio (HR) and 95% confidence interval (CI) denote the risk of death in the high expressing group compared to the low expressing group. The statistical significance (p-value) of each HR was determined using a Wald test. Median survival time is given in days with a 95% CI. (D) The relationship between miRNA expression and TNM stage was examined. Graphs show the proportion of individuals with high (red) and low (blue) miR-1, miR-133a, and miR-133b expression within each stage. Differences in TNM staging between high and low expressers was tested using Chi square tests, but there were no significant findings.
(TIFF)

**S6 Fig. Expression of miR-105 and miR-767 in head and neck, liver, lung, and stomach cancers.** (A) The distribution of miR-105-1, miR-105-2, and miR-767 in head and neck, liver, lung, and stomach cancer are presented such that miRNA expression is represented on the x-axis and density is represented on the y-axis. (B) Pairwise correlations between miR-105-1, miR-105-2, and miR-767 indicate concurrent expression of the three miRNA.
(TIFF)

**S7 Fig. Expression of miR-105 and miR-767 by sex.** (A) miR-105-1, (B) miR-105-2, and (C) miR-767 expression were compared between female (pink) and male (blue) patients in head and neck (H&N), liver, lung, and stomach cancers. For density plots (right), miRNA expression is represented on the x-axis and density is represented on the y-axis.
(TIFF)

**S8 Fig. miR-105 and miR-767 expression across cell lines.** (A) miR-105 and (B) miR-767 expression levels were measured across 13 cell lines. (C) Expression was consistent with patient data, as miR-105 and miR-767 expression were highly correlated. (D) miR-105 and (B) miR-767 expression was not correlated with miR-9 expression.
(TIFF)

**S9 Fig. miR-105 and miR-767 do not alter expression of AKT isoforms or PI3K subunits.** A549 cells were treated with a negative control miRNA (Control), miR-105 and miR-767 (miR), ZSTK474 alone, or ZSTK474 and miR-105/767 (ZSTK474 + miR). Gene expression was measured by qPCR. Each treatment condition included three replicates. ANOVA was performed. The p-values for the main effect of miRNA and ZSTK474 as well as their interaction are reported. Data are normalized to L7a and presented as mean ± SEM.
(TIFF)

**S1 Supplementary Methods. This section included additional methods in the study: Cell lines and treatments; miRNA and mRNA expression; Immunofluorescence; Cell viability assay; Flow cytometry; Western blot analysis; RNA immunoprecipitation (RIP).**
(DOCX)

## Acknowledgments

The authors acknowledge Dr Suparna Mantha of the Cancer Center at the Carle Foundation Hospital, for the helpful advice and constructive suggestions in conceptualizing the study. The authors thank the members of Pan and Chen laboratories for their technical support and helpful discussions throughout the study.

## Author Contributions

**Conceptualization:** Laura Moody, Yuan-Xiang Pan, Hong Chen.

**Data curation:** Laura Moody.

**Formal analysis:** Laura Moody.

**Funding acquisition:** Laura Moody, Yuan-Xiang Pan, Hong Chen.

**Investigation:** Laura Moody, Guanying Bianca Xu, Yuan-Xiang Pan, Hong Chen.

**Methodology:** Laura Moody, Hong Chen.

**Project administration:** Yuan-Xiang Pan, Hong Chen.

**Resources:** Yuan-Xiang Pan, Hong Chen.

**Software:** Laura Moody, Yuan-Xiang Pan.

**Supervision:** Yuan-Xiang Pan.

**Validation:** Laura Moody, Guanying Bianca Xu, Hong Chen.

**Visualization:** Laura Moody.

**Writing – original draft:** Laura Moody, Yuan-Xiang Pan.

**Writing – review & editing:** Laura Moody, Guanying Bianca Xu, Yuan-Xiang Pan, Hong Chen.

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
