## [Decision Letter · Decision Letter 0]

26 Jul 2021

Dear Dr. Chen,

Thank you very much for submitting your manuscript "Genome-wide cross-cancer analysis illustrates the critical role of bimodal miRNA in patient survival and drug responses" for consideration at PLOS Computational Biology.

As with all papers reviewed by the journal, your manuscript was reviewed by members of the editorial board and by several independent reviewers. In light of the reviews and the guest editor (below this email), we would like to invite the resubmission of a significantly-revised version that takes into account the reviewers' comments.

In addition, the guest editor also have some comments:

Line 68. The authors mentioned that mixture modeling has never been used to assess genome-wide bimodal microRNA expression. This claim is too strong. I encourage them to take a look at https://doi.org/10.1093/bioinformatics/btt599Besides bimodality, trimodality seems more reasonable since there could be low, no change, and high expression for a given microRNA w.r.t. to normal samples. Please explain.Line 85-89. The authors wrote "Bimodal miRNA were further confirmed by showing their usefulness in patient stratification based on overall survival and drug response (Figure 1B). After validating our methodology on breast cancer RNA-seq data". Validation results are not shown here. Figure 1B looks like just cartoon version of the workflow not the real results. Line 92: "We applied the MK approach and found that cancer tissue had more bimodal miRNA than control (Figure 2A)." How much of the difference is due to smaller sample size in normal tissues? Can authors run the same model on each cancer data using a bootstrap of the same sample size as the normal tissue for each cancer?Line 96-97: "At a bimodality index ... " Once again, how much is that due to sample size differences and batch effects rather than cancer types?Line 547: Supplementary Method file for model development and assessment is missing.

We cannot make any decision about publication until we have seen the revised manuscript and your response to the reviewers' comments. Your revised manuscript is also likely to be sent to reviewers for further evaluation.

Sincerely,

Yue Li, Ph.D.

Guest Editor

PLOS Computational Biology

Douglas Lauffenburger

Deputy Editor

PLOS Computational Biology

Reviewer's Responses to Questions

**Comments to the Authors:**

Reviewer #1: Manuscript entitled “Genome-wide cross-cancer analysis illustrates the critical role of bimodal miRNA in patient survival and drug responses” applied a mixture modeling approach to identify bimodal microRNA across cancers and showed the role of bimodal miRNA play in prognosis and drug response. Specifically, they identified two tumorigenic miRNA, miR-105 and miR767, which could be indicative of poor prognosis as well as involved in PI3K pathway.

This work first examined genome-wide bimodal expression across cancers using sequencing data and provides a framework for finding bimodal expression and demonstrates the role of bimodal miRNA in tumorigenesis as well as their potential in predicting patient survival and enabling effective treatment.

Overall, the authors present an interesting and well-written manuscript. Prior to publication, I would suggest consideration of the following concerns. These concerns stem from some questions I have when reading the paper. Clarifying them in the revised paper, I believe, will remove some of my confusions.

1. There are no descriptions of the parameters in Figure 1A.

2. In Figure 3A-H, there are no parameters described of the hierarchical clustering.

3. In Figure 3A-H, it would be great if you can switch group1 and group2 in some cancer types, for example, green for low expressed and purple for high expressed.

4. Considering the heterogeneity of cancers, shall we focus on cancer specific bimodal miRNA?

Reviewer #2: In this manuscript, Moody et al. used mixture models to analyze bimodal miRNA across cancer types, and identified a number of miRNAs that show bimodal patterns. They showed that overall, there are more bimodal miRNAs in cancer than in healthy tissues. They further described how these bimodal miRNAs, especially miR-105 and miR-767, can predict patient survival and drug response in some cancers.

Overall, I think the major mathematical framework for describing bimodality is logical and sound. The authors also performed extensive validations on a couple of key miRNAs from their analyses. However it is not convincing to me that these analyses and results support the main conclusions of the paper. The rationale and motivation for the study are also not as strong as I would expect.

Specific comments:

1. The abstract needs to be more coherent than the current version. The authors need to specify what motivates this work early in the abstract. Applying mixture models to study bimodal miRNA sounds like a logical step, but it is not clear to me what specific challenges they are trying to address and what they are hoping to find.

2. A main conclusion of this manuscript is that “Bimodal miRNA are ideal biomarkers that can be used to stratify patients for prognosis and drug response”. However, Figures 3 I-P show that for many cancer types, bimodal miRNAs do not stratify patients. And is the purple group or the green group supposed to survive longer? These two groups swap between different cancer types. Can the authors explain that?

3. Can the authors explain why they chose groups of bimodal miRNAs common to multiple cancer types to predict survival? They just mentioned that the estrogen receptor ESR1 and related miRNA may be bimodal in breast cancer, but they may not be bimodal in other cancer types.

4. Figure 2A: what bimodality index was used for this analysis? I feel the analysis in Figure 2B should be done here to assess whether the results are consistent when you vary the BI threshold.

Reviewer #3: The authors developed a new mixture modeling approach to identify bimodal miRNAs and then associate their profiles with cancer patient survival and drug response. The stratification of cancer patients based on bimodal miRNAs would be complementary to the stage-based stratification and would benefit the treatment of patients at different stages. It is quite impressive that the authors had tried to experimentally validate their discoveries/hypothesis regarding the role of bimodal miRNAs in facilitating response to drug treatment. Overall, it might not like a methodology oriented computational biology study, rather more like a bioinformatics-facilitated hypothesis-driven study. I have a few comments listed below.

Major:

1. The authors mentioned the novel model development and assessment was detailed in the supplementary method file. However, unfortunately I was not able to access to it – seems missing, which is indeed essential to evaluate the method contribution of this manuscript. Also, as a computational biology paper, I would suggest to move the essential contents of the novel method to the main-text and instead move lots of experimental details to the supplementary. Couple of the following comments are related to the missing of the supplementary method file, but nevertheless I list them separately.

2. With the emphasis of their novel method and the first study to analysis bimodal miRNA, I would suggest the authors to perform simulations to demonstrate the advance of their new method over others regarding identify bimodal miRNAs with higher sensitivity and lower false positive rate, etc.

3. Line 89, “After validating our methodology on breast cancer RNA-seq data,” – not clear to me whether the authors mentioned this methodology in the supplemental material? Or from their previous study (e.g. reference 7)? If the later, it will reduce the novelty of this manuscript regarding the new methodology although this is the first time applying on miRNA data.

4. It is not clear to me how many samples the authors used for assessing their models. I could guess the sample size from Figure 3, but please specify the details of analyzed samples in the genomic dataset section.

5. Line 555, the authors used the top 3,000 predicted targeted genes for both miR-105 and miR-767. It sounds too many targeted genes which might include lots of false positives. I’m curious were the nine target genes (line 189-190) at the top list of the predicted putative targets? Were these nine genes validated using qPCR (e.g. included in the 12 significantly down regulated)? I would suggest the authors to double check how many of these 3,000 genes are actually expressed in corresponded cancers using RNA-Seq data downloaded from GDCs. Alternatively, maybe roughly 500 to 1000 genes would be a good option?

6. Related to point 5, I’m curious that, rather than performing a multi-step investigations on identifying direct targets that inhibit drug effects, why not the authors go validation with the common nine targets or 36 putative targets of both miR-105 and miR-767 (indicated in line 189) and investigate their potential roles regarding drug response. Any comments?

7. What’s the insight of the single bimodal miRNAs? Would be false positives? Or have different patterns regarding patient stratification?

8. If I understand/interpret the results of Figure S7 correctly, the expression levels are quite low for both miR-105 and miR-767. Would these levels of expression be sufficient to reflect the drug response? Again - please add the sample size information for the investigation between miRNA and drug response.

9. Given PI3K, AKT and mTOR are highly connected, how to determine PI3K is the direct therapeutic target?

10. Line 255, any particular reasons or interpretations on why the %cells in early apoptosis starts to decrease at 0.8 uM while %cells in late apoptosis continue increasing.

Minor:

11. The resolution of the figures seems quite low even from the downloaded Tiff files as it look blurred when zooming in a bit. Please following the guideline of the journal and update the resolution.

12. Figure 7C, the y-axis go up to 60-100 fold changes which are extremely high. Would it be due to a very small expression level of the control condition?

13. Would the source codes become publically available (e.g. through GitHub)?

Reviewer #4: The authors utilized a mixture model to identify the miRNA with bimodal distribution among cancer patients across different types of cancer but not within the general healthy population. Further, they narrow down the cancer types and focus on two miRNAs, which show the potential as a prognosis marker and marker for the drug response. The authors did a lot of work to validate their computational predictions, which is fantastic from a point of view as a computational biologist. However, I have several concerns about the manuscript.

Major:

1. Supplementary method part is missing in the supplementary file, which makes the difficulty for the reviewer to judge their computational approach to calculate the bimodal index (in the first two paragraphs of the Method, they mentioned that the details will be described in supplementary method file…).

2. This is a comprehensive study, which would benefit more people if it is published in a general journal rather than a specialized computational journal… (especially, I am not sure how much innovation they made from the computational side due to the missing materials..)

3. How do they filter out the miRNA when the bimodal index is also high in general population? GTEx data is used? Again, this is not clear.

4. How do they determined the 1.4 cut off for bimodality index? It seems an arbitrary cut-off…

5. How do they identify the co-expression module of bimodal miRNA? This is important for the following study.

6. In Figure 4E, when cells were treated with miR-105 and miR-767 mimics, cell viability is increased. Will cell viability decrease when these two miRNA were inhibited?

Minor:

1. “in vitro” and “in vivo” should be italic.

2. The resolution of figure within the main pdf is really low…

3. In the discussion, “Previous studies have simply removed known bimodal genes from the analysis”., please add citation here.

**Have the authors made all data and (if applicable) computational code underlying the findings in their manuscript fully available?**

Reviewer #1: **No: **They haven't included them in the manuscript.

Reviewer #2: **No: **Not yet but they will make it available once accepted for publication.

Reviewer #3: **No: **The authors mentioned the detailed model development and assessment in the supplementary file - but seems missing. The authors also mentioned the file will be made public once accepted for publication but they didn't mention the source code - not sure whether the source code would be part of the supplementary file.

Reviewer #4: **No: **the method part is missing and the code is not shared

PLOS authors have the option to publish the peer review history of their article (what does this mean?). If published, this will include your full peer review and any attached files.

Reviewer #1: No

Reviewer #2: No

Reviewer #3: No

Reviewer #4: No
---

## [Decision Letter · Decision Letter 1]

12 Jan 2022

Dear Dr. Chen,

Thank you very much for submitting your manuscript "Genome-wide cross-cancer analysis illustrates the critical role of bimodal miRNA in patient survival and drug responses to PI3K inhibitors" for consideration at PLOS Computational Biology. As with all papers reviewed by the journal, your manuscript was reviewed by members of the editorial board and by several independent reviewers. The reviewers appreciated the attention to an important topic. Based on the reviews, we are likely to accept this manuscript for publication, providing that you modify the manuscript according to the review recommendations.

Sincerely,

Yue Li, Ph.D.

Guest Editor

PLOS Computational Biology

Douglas Lauffenburger

Deputy Editor

PLOS Computational Biology

[LINK]

Reviewer's Responses to Questions

**Comments to the Authors:**

Reviewer #1: Thank you for fully addressed my concerns, I have no more questions.

Reviewer #3: Thanks for the authors’ efforts. The authors have addressed most of my comments. I have a few minor comments:

1. After checking the detailed methodology, it seems the authors tried to control the false positives by adjusting the BI only through penalizing those also showing bimodal in controls. This is an interesting idea. The authors reported the top bimodal genes (Figure 2) and the total number of genes at different BI thresholds (Table 1) to show the advance of their new CMK method. However, I assume it would be more interesting to investigate examples that were detected by CMK but not MK to illustrate the contribution of introducing control samples. In addition, later on when applying the CMK on identifying bimodal miRNAs, the authors further emphasized to focus on those were common across cancers. I’m curious that whether or not considering the common bimodal miRNAs detected in cancers only would already compensate the risk of bimodal miRNAs in controls?

2. Line 119-120, the authors examined the known bimodal genes in breast cancer: ERS1, HER2 and PGR. Please add relevant references for these three bimodal genes.

3. Line 123, “which ranked it 21st (Table 1)”. However, BI rank of ESR1 for MK is 18 in Table 1. Please correct this typo.

4. The resolution of figures seems still low (i.e. 150 dpi).

Reviewer #4: the author has addressed all of my concerns.

**Have the authors made all data and (if applicable) computational code underlying the findings in their manuscript fully available?**

Reviewer #1: Yes

Reviewer #3: **No: **The authors provided a GitHub repository link for the source codes but currently unavailable to assess.

Reviewer #4: Yes

PLOS authors have the option to publish the peer review history of their article (what does this mean?). If published, this will include your full peer review and any attached files.

Reviewer #1: No

Reviewer #3: No

Reviewer #4: No

Figure Files:

Data Requirements:

Reproducibility:

References:

---

## [Decision Letter · Decision Letter 2]

15 Apr 2022

Dear Dr. Chen,

We are pleased to inform you that your manuscript 'Genome-wide cross-cancer analysis illustrates the critical role of bimodal miRNA in patient survival and drug responses to PI3K inhibitors' has been provisionally accepted for publication in PLOS Computational Biology.

Best regards,

Yue Li, Ph.D.

Guest Editor

PLOS Computational Biology

Douglas Lauffenburger

Deputy Editor

PLOS Computational Biology

Reviewer's Responses to Questions

**Comments to the Authors:**

Reviewer #1: The authors have addressed all my questions.

Reviewer #3: Thanks for the author's efforts. They have fully addressed all of my concerns.

Reviewer #4: the author has already addressed all my concerns in the last revision.

**Have the authors made all data and (if applicable) computational code underlying the findings in their manuscript fully available?**

Reviewer #1: Yes

Reviewer #3: Yes

Reviewer #4: None

PLOS authors have the option to publish the peer review history of their article (what does this mean?). If published, this will include your full peer review and any attached files.

Reviewer #1: No

Reviewer #3: No

Reviewer #4: No

---

## [Editor Report · Acceptance letter]

24 May 2022

PCOMPBIOL-D-21-00902R2 

Genome-wide cross-cancer analysis illustrates the critical role of bimodal miRNA in patient survival and drug responses to PI3K inhibitors

Dear Dr Chen,

I am pleased to inform you that your manuscript has been formally accepted for publication in PLOS Computational Biology. Your manuscript is now with our production department and you will be notified of the publication date in due course.

With kind regards,

Agnes Pap
